# UNCERTAINTY QUANTIFICATION FOR REGRESSION: A UNIFIED FRAMEWORK BASED ON KERNEL SCORES

## ABSTRACT

Regression tasks, notably in safety-critical domains, require proper uncertainty quantification, yet the literature remains largely classification-focused. In this light, we introduce a family of measures for total, aleatoric, and epistemic uncertainty based on proper scoring rules, with a particular emphasis on kernel scores. The framework unifies several well-known measures and provides a principled recipe for designing new ones whose behavior, such as tail sensitivity, robustness, and out-of-distribution responsiveness, is governed by the choice of kernel. We prove explicit correspondences between kernel-score characteristics and behavior of uncertainty measures, yielding concrete design guidelines for task-specific measures. Extensive experiments demonstrate that these measures are effective in downstream tasks and reveal clear trade-offs among instantiations, including robustness and out-of-distribution detection performance.

## 1 INTRODUCTION

Predictive models now drive decision-making in safety-critical domains such as weather forecasting (Price et al., 2025; Alet et al., 2025), autonomous driving (Michelmore et al., 2018) or healthcare (Löhr et al., 2024; Edupuganti et al., 2020); tasks where careful analysis of the model predictions and accurate uncertainty quantification are indispensable. Many studies have analyzed different approaches to quantify predictive uncertainty, often distinguishing between different sources of uncertainty. In particular, one usually considers two sources of uncertainty: *aleatoric uncertainty* and *epistemic uncertainty* (Hüllermeier & Waegeman, 2021). Broadly speaking, aleatoric uncertainty describes the inherent randomness in the data-generating process, for example, due to measurement errors and, as it describes variability that is independent of the amount of data, is often referred to as *irreducible* uncertainty. Epistemic uncertainty, on the other hand, arises from a lack of knowledge about the data-generating process and can be reduced by improving the model or acquiring more data; therefore, it is also referred to as *reducible* uncertainty.

While aleatoric uncertainty is well captured in predictive models, epistemic uncertainty is more difficult to represent and requires higher-order formalisms, such as second-order distributions (distributions of distributions). Following recent works (Hüllermeier & Waegeman, 2021; Kotelevskii et al., 2025), we refer to this as *uncertainty representation*. Given such a representation, the key question is how to measure or quantify the total, aleatoric, and epistemic uncertainty (*uncertainty quantification*). While the representation mainly determines performance, the choice of uncertainty measure plays a vital role in decision making and can have an additional impact on the performance of downstream tasks, with numerous works developing and analyzing new measures for uncertainty quantification (Sale et al., 2023a; Malinin & Gales, 2021; Kotelevskii et al., 2022; Berry & Meger, 2024). In addition, recent work focuses on steps towards more unified approaches that incorporate many existing measures and give guidance on how to construct new ones (Schweighofer et al., 2023; Kotelevskii et al., 2025). However, research has focused mainly on uncertainty quantification in classification, although many predictive models naturally operate in a regression setting.

In *supervised regression* tasks, a practitioner is generally interested in predictive uncertainty, which describes the uncertainty of the target $y \in \mathcal{Y}$ given some covariates $x \in \mathcal{X}$. While the notions of total, aleatoric, and epistemic uncertainty remain the same (Hüllermeier & Waegeman, 2021), the corresponding uncertainty measures fundamentally differ as compared to the classification case. Unlike classification, where the label space is discrete and bounded, regression targets lie in an

(often) unbounded, continuous and possibly high-dimensional domain, which often makes existing measures unsuitable. While in regression, many methods focus on uncertainty representation (Amini et al., 2020; Lakshminarayanan et al., 2017; Kelen et al., 2025), only a few works focus on analyzing the underlying uncertainty measures (Berry & Meger, 2024; Bülte et al., 2025b).

**Contributions** In this paper, we introduce a unified framework for uncertainty quantification in multivariate regression, built from proper scoring rules. Following Kotelevskii et al. (2025); Hofman et al. (2024b), we formulate uncertainty measures in terms of score (or Bregman) divergences. Similar to Gruber & Buettner (2024), we propose to use *kernel scores* (Gneiting & Raftery, 2007) as a specific instantiation for the uncertainty measures, as those offer unique advantages as compared to other scoring rules (Waghmare & Ziegel, 2025) and establish new connections to proper scoring rules in real-valued domains. We not only show that this framework includes several already existing uncertainty measures, but also provide a principled way to design new uncertainty measures based on corresponding properties of the underlying kernel score. We derive explicit connections between those properties and desirable behavior of the associated uncertainty measure, such as translation invariance or robustness. Finally, we validate the proposed measures empirically, highlighting the derived theoretical properties in practice and showcasing their application in several downstream decision-making tasks.

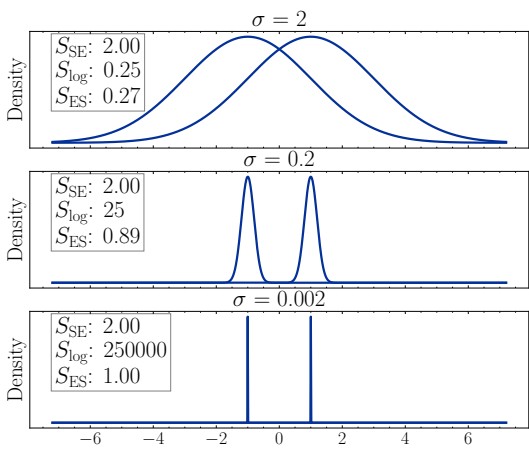

Figure 1: Illustration of epistemic uncertainty for a two-member Gaussian ensemble with shared variances. As the component variances shrink, the variance-based measure ($S_{\text{SE}}$) stays constant, the entropy-based measure ($S_{\log}$) diverges, while our proposed energy-score-based measure ($S_{\text{ES}}$) converges to half the Euclidean distance between component means.

## 2 UNCERTAINTY IN SUPERVISED REGRESSION

In the following, we denote by $\mathcal{X} \subseteq \mathbb{R}^d$ and $\mathcal{Y} \subseteq \mathbb{R}^d$ the (real-valued) feature and target space, respectively. Furthermore, let $\sigma(\mathcal{Y})$ be the Borel $\sigma$-algebra on $\mathcal{Y}$, let $\mathcal{P}$ denote a convex set of probability measures on the measure space $(\mathcal{Y}, \sigma(\mathcal{Y}))$ and let $\overline{\mathbb{R}} = \mathbb{R} \cup \{-\infty, \infty\}$. In addition, we write $\mathcal{D} = \{\boldsymbol{x}_i, \boldsymbol{y}_i\}_{i=1}^n \in (\mathcal{X} \times \mathcal{Y})^n$ for the training data. For $i \in \{1, \ldots, n\}$, each pair $(\boldsymbol{x}_i, \boldsymbol{y}_i)$ is a realization of the random variables $(X_i, Y_i)$, which are assumed to be independent and identically distributed (i.i.d) according to a probability measure $\mathbb{P}$. Consequently, each $\boldsymbol{x} \in \mathcal{X}$ induces a conditional probability distribution $\mathbb{P}(\cdot \mid \boldsymbol{x})$, where $\mathbb{P}(\boldsymbol{y} \mid \boldsymbol{x})$ represents the probability of observing the outcome $\boldsymbol{y} \in \mathcal{Y}$ given the features $\boldsymbol{x}$. Here, we assume that the conditional predictive distribution $\mathbb{P}(\cdot \mid \boldsymbol{x})$ is absolutely continuous with respect to the Lebesgue measure $\mu$ and therefore admits a probability density function $p(\cdot \mid \boldsymbol{x})$.

### 2.1 UNCERTAINTY REPRESENTATION

Regarding second-order uncertainty quantification, we denote by $\mathcal{P}(\mathcal{Y})$ the set of all (convex) probability measures on $\mathcal{Y}$ on the measurable space $(\mathcal{Y}, \sigma(\mathcal{Y}))$ and, similarly, by $\mathcal{P}(\mathcal{P}(\mathcal{Y}))$ the set of all probability measures on $(\mathcal{P}(\mathcal{Y}), \sigma(\mathcal{P}(\mathcal{Y})))$. We refer to $Q \in \mathcal{P}(\mathcal{P}(\mathcal{Y}))$ as a *second-order distribution*. In contrast to the classification setting, the probability measures $\mathbb{P} \in \mathcal{P}(\mathcal{Y})$, are not necessarily defined on a bounded domain. While we keep the setup as general as possible and this article mainly revolves around uncertainty quantification rather than uncertainty representation, the following examples illustrate how a second-order distribution could be specified within our framework:

*Parametric distributions:* Given a (fixed) parametric distribution $p(\boldsymbol{y} \mid \boldsymbol{\theta}(\boldsymbol{x}))$ with $\boldsymbol{\theta} \in \Theta \subseteq \mathbb{R}^p$, we can consider the second-order distribution to be on the (measurable) parameter space $(\Theta, \sigma(\Theta))$,

e.g. $Q \in \mathcal{P}(\Theta)$. In particular, this includes many uncertainty quantification methods, such as deep ensembles (Lakshminarayanan et al., 2017), deep evidential regression (Amini et al., 2020), or distributional regression (Kneib et al., 2023).

*Ensemble approaches:* Given an empirical measure, i.e. $Q = Q_m = \frac{1}{M} \sum_{m=1}^{M} \delta_{\mathbb{P}_m}$ for first-order distributions $\mathbb{P}_m \sim Q$, the setting includes general ensemble approaches, such as ensembles of normalizing flows (Berry & Meger, 2023), mixture density networks (Bishop, 1994), nonparametric ensembles (Kelen et al., 2025) or diffusion models (Wolleb et al., 2021).

Unless noted otherwise, we will consider arbitrary first- and second-order distributions, where we assume that we have a first-order distribution $\mathbb{P} \sim Q$, distributed to some second-order distribution $Q$ and $Y \sim \mathbb{P}$. In addition, we define the first-order probability measure $\overline{\mathbb{P}} := \mathbb{E}_Q[\mathbb{P}]$, which can be interpreted as the Bayesian model average (BMA) predictive distribution (Schweighofer et al., 2023).

## 3 UNCERTAINTY QUANTIFICATION BASED ON PROPER SCORING RULES

In this section, we present a general framework for (second-order) uncertainty quantification based on proper scoring rules, enabling a unified theoretical treatment of different uncertainty measures. A *scoring rule* is a function $S : \mathcal{P} \times \mathcal{Y} \to \overline{\mathbb{R}}$, such that $S(\mathbb{P}, \mathbb{Q}) := \int S(\mathbb{P}, \boldsymbol{y}) \, d\mathbb{Q}(\boldsymbol{y})$ is well-defined for all $\mathbb{P}, \mathbb{Q} \in \mathcal{P}$ (Gneiting & Raftery, 2007). A scoring rule $S$ is called *proper*, if

$$S(\mathbb{Q}, \mathbb{Q}) \leq S(\mathbb{P}, \mathbb{Q}), \quad \text{for all } \mathbb{P}, \mathbb{Q} \in \mathcal{P} \tag{1}$$

and *strictly proper* if equality holds only when $\mathbb{P} = \mathbb{Q}$. Intuitively, proper scoring rules quantify the discrepancy between a predictive distribution and the realized outcome, attaining their minimum at the true distribution. Following Dawid (2007), every scoring rule $S$ can be associated with a *(generalized) entropy $H$* and a *divergence $D$*, via

$$H : \mathcal{P} \to \overline{\mathbb{R}}, \qquad \mathbb{P} \mapsto H(\mathbb{P}) := S(\mathbb{P}, \mathbb{P}) \tag{2}$$

$$D : \mathcal{P} \times \mathcal{P} \to \overline{\mathbb{R}}, \quad (\mathbb{P}, \mathbb{Q}) \mapsto D(\mathbb{P}, \mathbb{Q}) := S(\mathbb{P}, \mathbb{Q}) - H(\mathbb{Q}). \tag{3}$$

For (strictly) proper scoring rules, $H$ is (strictly) concave on $\mathcal{P}$, while the divergence satisfies $D(\mathbb{P}, \mathbb{Q}) \geq 0$ for $\mathbb{P}, \mathbb{Q} \in \mathcal{P}$ with equality if and only if $\mathbb{P} = \mathbb{Q}$ (compare Dawid, 2007). These quantities generalize the familiar notions of Shannon entropy and Kullback-Leibler divergence: $H$ captures the average surprisal under a distribution, and $D$ measures the discrepancy between two distributions. Under mild assumptions, proper scoring rules can be characterized in terms of their entropy function (Gneiting & Raftery, 2007), so either can be used to construct the other.

Building on the above, we define the following estimator (Kotelevskii et al., 2025; Hofman et al., 2024b)

$$\text{TU}_{\text{B}}(Q) := \mathbb{E}_{\mathbb{P} \sim Q}[S(\overline{\mathbb{P}}, \mathbb{P})], \quad \text{EU}_{\text{B}}(Q) := \mathbb{E}_{\mathbb{P} \sim Q}[D(\overline{\mathbb{P}}, \mathbb{P})], \quad \text{AU}_{\text{B}}(Q) := \mathbb{E}_{\mathbb{P} \sim Q}[H(\mathbb{P})], \tag{4}$$

which is based on the BMA predictive distribution and recovers variance- and entropy-based measures as special cases. However, since the BMA distribution might differ from the true predictive distribution, for example in a misspecified model, this estimator can be misleading (Schweighofer et al., 2023). Recent work (Kotelevskii et al., 2025; Schweighofer et al., 2023) therefore considers *pairwise comparisons* between predictive distributions of all models weighted by their posterior probabilities, yielding

$$\text{TU}_{\text{P}}(Q) := \mathbb{E}_{\mathbb{P}, \mathbb{P}' \sim Q}[S(\mathbb{P}', \mathbb{P})], \quad \text{EU}_{\text{P}}(Q) := \mathbb{E}_{\mathbb{P}, \mathbb{P}' \sim Q}[D(\mathbb{P}', \mathbb{P})], \quad \text{AU}_{\text{P}}(Q) := \mathbb{E}_{\mathbb{P} \sim Q}[H(\mathbb{P})]. \tag{5}$$

Here, AU remains unchanged, while TU and EU are defined relative to the true belief $Q$. Both estimators satisfy the additive decomposition $\text{TU} = \text{EU} + \text{AU}$. While the pairwise estimator (P), as opposed to the BMA estimator (B), admits closed-form solutions for many distributions, it comes at higher computational cost, for example $\mathcal{O}(M^2)$ vs. $\mathcal{O}(M)$ for a second-order ensemble of size $M$.

Comparing both estimators, the difference

$$\Delta := \text{TU}_{\text{P}} - \text{TU}_{\text{B}} = \text{EU}_{\text{P}} - \text{EU}_{\text{B}} = \mathbb{E}_{\mathbb{P} \sim Q}[\mathbb{E}_{\mathbb{P}' \sim Q}[S(\mathbb{P}', \mathbb{P})] - S(\overline{\mathbb{P}}, \mathbb{P})], \tag{6}$$

quantifies how the BMA score deviates from the expected score over all models. If $S$ is convex in its first argument, Jensen's inequality implies $\Delta \geq 0$, therefore the pairwise estimator is an upper bound for the BMA estimator (Schweighofer et al., 2023). From now on, we refer to the two different methods with index *B* and *P* for BMA and pairwise estimation, respectively.

## 4 KERNEL SCORES

In order to guide the choice of $S$ for the instantiations of uncertainty estimates, we now introduce an important subclass of scoring rules, so-called kernel scores, which have many favorable properties and are widely studied in the machine learning literature. Kernel scores have been first discussed by Dawid (2007); Gneiting & Raftery (2007); here we draw mainly on the notation from Waghmare & Ziegel (2025). Consider a continuous, negative definite kernel[1] $k$, denote $\mathcal{P}_k = \{\mathbb{P} \in \mathcal{P} : \iint k(\boldsymbol{x}, \boldsymbol{x}')\, d\mathbb{P}(\boldsymbol{x})\, d\mathbb{P}(\boldsymbol{x}') < \infty\}$, and, without loss of generality, assume that $k(\boldsymbol{x}, \boldsymbol{y}) \geq 0, \ \forall \boldsymbol{x}, \boldsymbol{y} \in \mathcal{Y}$.

**Definition 4.1** (Kernel score). *The kernel score $S_k : \mathcal{P}_k \times \mathcal{Y} \mapsto \overline{\mathbb{R}}$ associated with the kernel $k : \mathcal{Y} \times \mathcal{Y} \to [0, \infty)$ is defined as*

$$S_k(\mathbb{P}, \boldsymbol{y}) = \int k(\boldsymbol{x}, \boldsymbol{y})\, d\mathbb{P}(x) - \frac{1}{2} \iint k(\boldsymbol{x}, \boldsymbol{x}')\, d\mathbb{P}(\boldsymbol{x})\, d\mathbb{P}(\boldsymbol{x}') - \frac{1}{2} k(\boldsymbol{y}, \boldsymbol{y}), \tag{7}$$

*for $\mathbb{P} \in \mathcal{P}, \boldsymbol{y} \in \mathcal{Y}$.*

This scoring rule is (strictly) proper for a (strongly) conditionally negative definite kernel (Waghmare & Ziegel, 2025). Similar to Ziegel et al. (2024) we include the last term in the above definition, which ensures that the kernel score $S_k$ is nonnegative. The entropy and divergence associated with a kernel score $S_k$ and $\mathbb{P}, \mathbb{Q} \in \mathcal{P}_k$ are given as

$$H_k(\mathbb{P}) = \frac{1}{2} \iint k(\boldsymbol{x}, \boldsymbol{x}')\, d\mathbb{P}(\boldsymbol{x})\, d\mathbb{P}(\boldsymbol{x}') - \frac{1}{2} \int k(\boldsymbol{x}, \boldsymbol{x})\, d\mathbb{P}(\boldsymbol{x}), \tag{8}$$

$$D_k(\mathbb{P}, \mathbb{Q}) = -\frac{1}{2} \iint k(\boldsymbol{y}, \boldsymbol{y}')\, d(\mathbb{P} - \mathbb{Q})(\boldsymbol{y})\, d(\mathbb{P} - \mathbb{Q})(\boldsymbol{y}'). \tag{9}$$

For the kernel score, the corresponding divergence $D_k$ recovers the squared Maximum Mean Discrepancy (MMD[2]) (Gretton et al., 2012), which plays an important role in statistics and machine learning (Gretton et al., 2012; Sejdinovic et al., 2013). In fact, kernel scores admit many advantageous properties:

*Metric on $\mathcal{P}_k$:* Under mild conditions, kernel scores are the only scoring rules that are a valid metric on $\mathcal{P}_k$ (Theorem 19, Waghmare & Ziegel, 2025). Furthermore, the only restriction on the existence of the score (and divergence) is that $H_k(\mathbb{P}) < \infty$, as by definition of $\mathcal{P}_k$. In particular, this allows for measuring the divergence between continuous, discrete, or even degenerate distributions, as opposed to other scoring rules that require absolute continuity with respect to the Lebesgue measure (compare Figure 1).

*Flexible choice of $k$:* The general definition of the kernel score in (7) allows for a broad choice of underlying domains. While in this article, we focus on uni or multivariate regression, many kernels have been developed for other domains. This includes, in particular, kernels for spatial data (Scheuerer & Hamill, 2015), graph data (Vishwanathan et al., 2010), functional data (Wynne & Duncan, 2022), or natural language (Lodhi et al., 2002).

*Unbiased estimation:* The MMD[2] (and therefore also $S_k$ and $H_k$) admit an unbiased empirical estimator (Gretton et al., 2012). Therefore, it can be used as a measure even if no closed-forms are available, as opposed to, for example, the log-score, which does not admit an unbiased estimator (Paninski, 2003).

*Translation invariance:* Kernel scores with a kernel of the form $k(\boldsymbol{x}, \boldsymbol{y}) \equiv k(\boldsymbol{x} - \boldsymbol{y})$, $\boldsymbol{x}, \boldsymbol{y} \in \mathcal{Y}$ are *translation invariant* in the sense that $S_k(\mathbb{P}, \boldsymbol{y}) = S_k(\mathbb{P}_{\boldsymbol{h}}, \boldsymbol{y} + \boldsymbol{h})$ for $\boldsymbol{y}, \boldsymbol{h} \in \mathcal{Y}$, where $\mathbb{P}_{\boldsymbol{h}}(A) = \mathbb{P}(A + \boldsymbol{h})$ for Borel sets $A \subseteq \mathcal{Y}$ (Waghmare & Ziegel, 2025).

*Homogeneity:* A scoring rule $S$ is said to be *homogeneous* of degree $\alpha$ if $S(\mathbb{P}_c, cy) = c^\alpha S(\mathbb{P}, y)$ for every $c > 0, \mathbb{P} \in \mathcal{P}$ and $y \in \mathcal{Y}$, where $\mathbb{P}_c(A) = \mathbb{P}(c^{-1}A)$ for Borel sets $A \subseteq \mathcal{Y}$. Thus, affine transformations of the data distribution lead to the same performance assessment of the scoring rule (or scaled by a factor $\alpha$).

---

[1] A kernel $k : \mathcal{Y} \times \mathcal{Y} \to \mathbb{R}$ is negative definite if $\sum_{i,j=1} a_i a_j k(x_i, x_j) \leq 0, \ \forall n \in \mathbb{N}, \ a_1, \ldots, a_n \in \mathbb{R}, \ x_1, \ldots, x_n \in \mathcal{Y}$ (Waghmare & Ziegel, 2025).

## 5 PROPERTIES OF KERNEL SCORES AS AN UNCERTAINTY MEASURE

We now want to analyze the properties of the uncertainty measures in (4) and (5) if they are instantiated with a (proper) kernel score $S_k$. It is noteworthy that the characteristics of the kernel scores, introduced in the previous section, directly transfer to the corresponding uncertainty measures. Furthermore, depending on the task, these properties can be very important in the context of uncertainty quantification. For instance, kernel scores allow for comparing (almost any) arbitrary distributions with an unbiased estimator, which can be important, for example when different first-order distributions are combined via a linear pool (Hall & Mitchell, 2007) or in forecast surveys (Krüger & Nolte, 2016). In addition, we show that, if choosing $k$ in a principled way, the uncertainty measures instantiated with $S_k$ fulfill intuitive properties that have been studied in the literature (Wimmer et al., 2023; Sale et al., 2023a; Bülte et al., 2025b). One trivial aspect of the corresponding measures is that they are all nonnegative, which follows directly from the kernel being nonnegative. In addition, we show that, under some assumptions on $S_k$, the measures assign higher values for EU (or AU) if the corresponding second-order (first-order) distribution has higher variability. Finally, we analyze the robustness of the corresponding uncertainty measures with respect to a perturbation in the second-order distribution.

Before we show the corresponding results, we need to introduce some notation. Let $\mathbb{P} \sim Q, \mathbb{P}' \sim Q'$ be two random first-order distributions with $Q, Q' \in \mathcal{P}(\mathcal{P}(\mathcal{Y}))$. Furthermore, let $\delta_{\mathbb{P}} \in \mathcal{P}(\mathcal{P}(\mathcal{Y}))$ denote the Dirac measure at $\mathbb{P} \in \mathcal{P}(\mathcal{Y})$. For $\mathbb{P}_1, \mathbb{P}_2 \in \mathcal{P}(\mathcal{Y})$ let $\leq_{\mathrm{cx}}$ denote the convex order, meaning that $\mathbb{P}_1 \leq_{\mathrm{cx}} \mathbb{P}_2 \iff \mathbb{E}_{X \sim \mathbb{P}_1}[\phi(X)] \leq \mathbb{E}_{Y \sim \mathbb{P}_2}[\phi(Y)]$ for all convex functions $\phi : \mathcal{Y} \to \mathbb{R}$. Similarly, for $Q_1, Q_2 \in \mathcal{P}(\mathcal{P}(\mathcal{Y}))$, let $\leq_{\mathrm{cx}}^2$ denote the convex order with respect to all convex functionals $\Phi : \mathcal{P}(\mathcal{Y}) \to \mathbb{R}$. In particular for $\mathbb{P}_1 \leq_{\mathrm{cx}} \mathbb{P}_2$ it holds that $\mathbb{E}_{X \sim \mathbb{P}_1}[X] = \mathbb{E}_{Y \sim \mathbb{P}_2}[Y]$ and $\mathbb{V}_{X \sim \mathbb{P}_1}[X] \leq \mathbb{V}_{Y \sim \mathbb{P}_2}[Y]$, since the stochastic order is a measure of variability of a distribution (Shaked & Shanthikumar, 2007). Then, we obtain the following properties of the corresponding uncertainty measures[2], which are proved in Appendix A.

**Proposition 5.1.** *For any proper scoring rule $S$, it holds that*

1. *$Q = \delta_{\mathbb{P}} \implies \mathrm{EU}(Q) = 0$, while for a strictly proper scoring rule the converse holds as well,*

2. *$\mathrm{EU}(\delta_{\mathbb{P}}) \leq \mathrm{EU}(Q_1) \leq \mathrm{EU}(Q_2), \quad \forall Q_1 \leq_{\mathrm{cx}}^2 Q_2$.*

Intuitively, since $Q_1$ has less variability than $Q_2$, the corresponding measure of epistemic uncertainty assigns a smaller value to $Q_1$ as well. Consequently, the smallest value of EU should be attained for a distribution with no variability at all, which is in the case of a (second-order) Dirac distribution $\delta_{\mathbb{P}}$. In addition, the converse holds for a strictly proper scoring rule, which means that $\mathrm{EU}(Q) = 0$ can only be attained for the Dirac distribution $Q = \delta_{\mathbb{P}}$. Wimmer et al. (2023); Sale et al. (2023a) formulate similar arguments for a mean-preserving spread in the classification case. We transfer this to the regression case and reformulate it using the convex order.

**Proposition 5.2.** *Any kernel score $S_k$ with a translation invariant kernel $k(x, x')$ that is convex in one of its arguments fulfills $\mathrm{AU}(\delta_{\mathbb{P}_1}) \leq \mathrm{AU}(\delta_{\mathbb{P}_2}), \forall \mathbb{P}_1 \leq_{\mathrm{cx}} \mathbb{P}_2$.*

Similar to 5.1, if the first-order distribution $\mathbb{P}_1$ has less variability than $\mathbb{P}_2$, the corresponding measure of AU is smaller as well. Again, this is similar to studied properties in the classification case (Wimmer et al., 2023; Sale et al., 2023a).

**Proposition 5.3.** *Consider a parametric first-order distributions $\mathbb{P}_{\boldsymbol{\theta}} \in \mathcal{P}(\mathcal{Y})$ with $\boldsymbol{\theta} \in \Theta \subseteq \mathbb{R}^p$, a corresponding second-order distributions $Q \in \mathcal{P}(\Theta)$, first-order distribution $\boldsymbol{\vartheta} \sim Q$ and assume that $\mathrm{AU}(Q) < \infty$. Furthermore, define $Q_\varepsilon := (1 - \varepsilon)Q + \varepsilon\delta_{\boldsymbol{\theta}_0}, \boldsymbol{\theta}_0 \in \Theta$ and consider the influence function (IF):*

$$\mathrm{IF}(\boldsymbol{\theta}_0; \mathrm{AU}, Q) = \lim_{\varepsilon \to 0} \frac{\mathrm{AU}(Q_\varepsilon) - \mathrm{AU}(Q)}{\varepsilon} = H_k(\mathbb{P}_{\boldsymbol{\theta}_0}) - \mathbb{E}_Q[H_k(\mathbb{P}_{\boldsymbol{\vartheta}})].$$

*We then have that any kernel score $S_k$ with bounded kernel $k$ is robust in terms of the influence function.*

---

[2]Since propositions 5.1-5.3 hold for both type of estimators, we do not use an index B/P here.

This definition of robustness of an estimator via the influence function (Hampel et al., 1986, Chapter 2), analyzes the limiting behavior if the underlying (second-order) distribution is perturbed by a single point diverging to infinity. If the influence function is bounded, any outlier in $Q$ can only have finite impact on the estimation of $\mathrm{AU}(Q)$, making it robust against such outliers. While the influence function could in principle also be defined for arbitrary second-order distributions, it is not straightforward to define the contamination $Q_\epsilon$ and the corresponding convergence for arbitrary measures.

Based on the previous propositions, one can choose different instantiations of the uncertainty measures, based on different choices of the kernel function $k$. In particular, we propose the following choice of kernels, which might be selected based on the underlying task. The corresponding derivations can be found in Appendix B.

*Squared-error:* When choosing $k(x, x') = \|x - x'\|^2$ we obtain the squared error $S_{\mathrm{SE}}$, which, in the univariate case, leads to the commonly-used variance-based measure. It fulfills (5.2), but not (5.1), since the corresponding scoring rule is not strictly proper.

*Energy score:* When $k(x, x') = \|x - x'\|^\beta$, $\beta \in (0, 2)$, we obtain the (strictly proper) energy score $S_{\mathrm{ES}}$ (Gneiting & Raftery, 2007) and the corresponding divergence, the *energy distance* (Székely & Rizzo, 2013). A special case of the former is the continuous ranked probability score (CRPS) (Gneiting & Raftery, 2007), which arises for $d = 1, \beta = 1$. It is the only homogeneous translation invariant kernel score on $\mathbb{R}^d$ (Waghmare & Ziegel, 2025) and fulfills (5.1) and (5.2).

*Gaussian kernel score:* Another important example arises when we choose $k$ as the Gaussian kernel $k(x, x') = -\exp\left(-\|x - x'\|^2/\gamma^2\right)$ with bandwidth $\gamma$, which is also a strictly proper scoring rule (denoted as $S_{k_\gamma}$) and therefore fulfills (5.1). In addition, it is robust and is the only proposed score that fulfills (5.3), as the corresponding kernel is bounded.

While the well-known *log-score* $S_{\log}$, which corresponds to the entropy-based measure, is also a scoring rule, it is not a kernel score. In particular, it can be negative and therefore difficult to interpret. However, it still fulfills (5.1) and (5.2) under some assumptions, as shown in Appendix A. Each of the kernel scores mentioned above, as well as their corresponding uncertainty measure, can be suitable for uncertainty quantification, depending on the underlying task. For example, when mainly interested in the location estimate of a distribution, the squared error might be suitable, as it measures EU only in the first moments of the first-order distributions. On the other hand, for spatial data, the energy score might be more appropriate, as it is translation-invariant and homogeneous.

## 6 NUMERICAL EXPERIMENTS

In this section, we provide several numerical experiments that highlight differences and similarities of the corresponding kernel instantiations and highlight their applicability as uncertainty measures and in downstream tasks. In general, the evaluation of (second-order) uncertainty measures is not straightforward, as no ground truth uncertainty is available. While the evaluation focuses mainly on the properties and instantiations of the aforementioned kernel scores, we also include the log-score as a comparison, since it is commonly used in practice to assess uncertainty. While the Gaussian kernel score $S_{k_\gamma}$ requires tuning of the bandwidth, we found that choosing $\gamma$ with the median heuristic works well empirically and use that value for all experiments. However, we provide an ablation study on the choice of the bandwidth in Appendix C. In the following, we use the pairwise uncertainty measures, as closed-form expressions are available for different first-order distributions (compare Appendix B). More details on each experiment can be found in Appendix C.

### 6.1 OUT-OF-DISTRIBUTION DETECTION

First, we use a distributional regression network (DRN) (Rasp & Lerch, 2018) to predict the 2-meter surface temperature (T2M) across Europe. The DRN gets a numerical weather prediction as the input and predicts a Gaussian distribution $\mathcal{N}_{\mu_{l,t}, \sigma_{l,t}^2}$, where $t$ denotes the time and $l$ is an index for the gridpoint. We follow the setup in Bülte et al. (2025a) and train an ensemble of $M = 10$ DRNs, as well as a deep evidential regression (DER) model, solely on gridpoints over land, but evaluate over the whole domain, allowing for assessing the performance on out-of-distribution data. As the predictability of the surface temperature changes with altitude, one would additionally expect

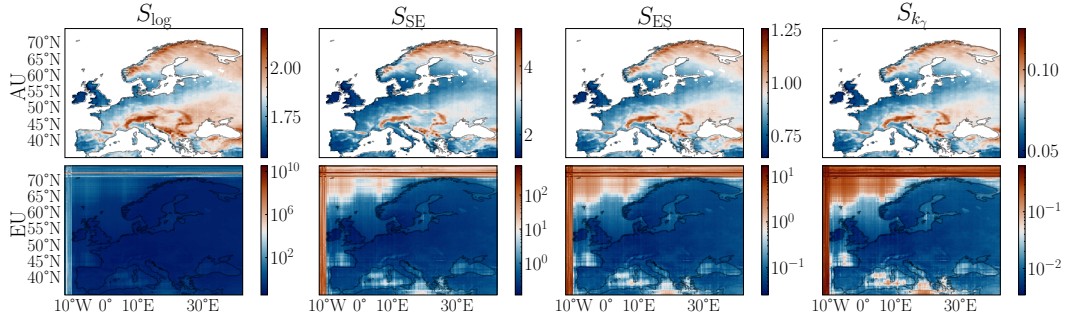

Figure 2: The figure shows AU and EU averaged over a test set of 365 days for the different uncertainty measures. For visualization purposes, epistemic uncertainty is shown on a log-scale and the land-sea mask is applied to aleatoric uncertainty, as it does not affect out-of-distribution performance.

aleatoric uncertainty to change with the orography, while epistemic uncertainty should change with the land-sea mask (both visualized in Appendix C).

Figure 2 shows the aleatoric and epistemic uncertainty for all measures, averaged across the test data. For AU all measures produce similar uncertainties, although, as opposed to the kernel scores, $S_{\log}$ shows high values of AU for larger areas of the domain. A similar visualization for deep evidential regression can be found in Appendix C.

Table 1: Results for out-of-distribution detection (AUROC ↑) for the different uncertainty measures, representation methods, and datasets. The best result is underlined, all within its standard deviation in bold. $S_{\log}$ is not available for the sample-based methods.

| Experiment | Method | $S_{\log}$ | $S_{\mathrm{SE}}$ | $S_{\mathrm{ES}}$ | $S_{k_\gamma}$ |
|---|---|---|---|---|---|
| **T2M** | DRN | **0.856** | 0.847 | 0.854 | 0.853 |
| | DER | **0.524** | 0.440 | 0.458 | 0.451 |
| **ApolloScape** | DER | 0.970 | 0.989 | **0.997** | 0.994 |
| | CRPS | - | 0.528 | **0.577** | 0.533 |
| | MvNormal | - | 0.545 | **0.566** | 0.551 |
| Average rank | | 2.00 | 3.40 | **1.40** | 2.40 |

To assess epistemic uncertainty and out-of-distribution detection performance of the different measures, we calculate the AUROC of the uncertainty scores on ID (land) and OOD (sea) data. Here, $S_{\log}$ performs best for both methods, while generally the DER model leads to significantly worse performance, as highlighted in Table 1.

Similarly, we use the depth regression setup from Amini et al. (2020) to further assess OOD detection performance. Models are trained using an ID (indoor scenes) and OOD (driving scenes) dataset of image-depth pairs. As uncertainty representation methods, we use deep evidential regression Amini et al. (2020), the CRPS method by Kelen et al. (2025) and a multivariate Gaussian regressor (MvNormal) (Muschinski et al., 2024). The second-order distribution for the latter two methods is generated by MCDropout (Gal & Ghahramani, 2016). For more details compare Appendix C.

Table 1 includes the corresponding results, highlighting that the measure based on the energy score leads to the best performance across all representation methods. Again, DER performs best, possibly because MCDropout is not perfectly suited to represent a second-order distribution.

## 6.2 ROBUSTNESS ANALYSIS

In order to empirically validate the robustness (in terms of the influence function) of different measures, we use three datasets from the UCI benchmark (Hernández-Lobato & Adams, 2015) and train a deep ensemble (Lakshminarayanan et al., 2017) on each task. Then, we train one additional en-

Table 2: The table shows the mean absolute percentage error ($\downarrow$) of aleatoric and epistemic uncertainty for $M = 25$ ensemble members and one additional ensemble member with target distortion $\delta$.

| $S/\delta$ | Aleatoric | | | | Epistemic | | | |
|---|---|---|---|---|---|---|---|---|
| | 0.0 | 1.0 | 2.5 | 5.0 | 0.0 | 1.0 | 2.5 | 5.0 |
| $S_{\log}$ | 0.25 | 3.1 | 4.5 | 4.8 | 3.5 | 1.5e+04 | 5.3e+04 | 2.4e+05 |
| $S_{\mathrm{SE}}$ | 1.1 | 4.6e+03 | 6.8e+04 | 4.8e+05 | 3.7 | 3e+04 | 6.2e+04 | 4.7e+05 |
| $S_{\mathrm{ES}}$ | 0.55 | 67 | 2.2e+02 | 5e+02 | 2.7 | 8.1e+02 | 1.2e+03 | 4.1e+03 |
| $S_{k_\gamma}$ | 0.027 | 0.17 | 0.19 | 0.19 | 1.6 | 13 | 12 | 13 |

semble member using a target variable with added noise, i.e. $\tilde{y} = y + \mathcal{N}(0, \delta^2)$ with gradually increasing noise. This allows for comparing the robustness of the different measures with respect to an outlier in the second-order distribution. To measure the deviation, we use the mean absolute percentage error (MAPE) with respect to the uncertainty in the base ensemble, i.e.,

$$\mathrm{MAPE} \coloneqq \frac{100}{n} \sum_{i=1}^{n} \left| \frac{M_i^\delta - M_i}{M_i} \right|,$$

where $M_i \in \{\mathrm{AU}, \mathrm{EU}\}$ is the measure of uncertainty for samples $i = 1, \ldots, n$ and $M_i^\delta$ denotes the measure of uncertainty for the distorted prediction. Table 2 shows the results for the concrete dataset. Due to its robustness, the Gaussian kernel score changes the least, while the variance-based measure quickly diverges to extreme values. More detailed results and a theoretical analysis of robustness for deep ensembles can be found in Appendix C.

### 6.3 TASK ADAPTATION OF MEASURES

Recent work suggests that there is no universally optimal uncertainty measure (Mucsányi et al., 2024), which motivates us to analyze how uncertainty measures can be adapted and tailored to specific tasks. Even within the kernel score framework, a wide range of measures can be constructed by choosing different kernels $k$. Each kernel choice not only defines an uncertainty measure, but also induces a corresponding training/validation loss via its scoring rule $S_k$, referred to as task loss. Understanding the relationship between a task loss and the associated uncertainty measures is therefore central to task adaption.

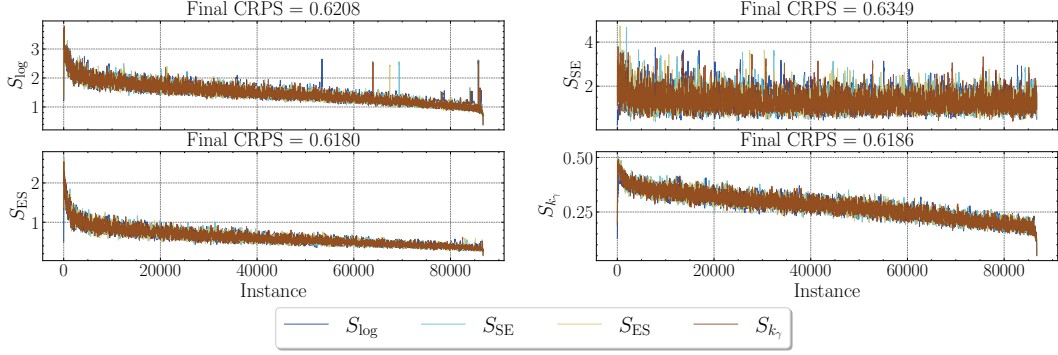

Figure 3: Different task losses (each plot) sorted by each of the different uncertainty measures from highest to lowest total uncertainty, trained on the T2M prediction task. For visualization purposes, the values shown are moving averages of size 50.

We first investigate this connection using the task of post-processing 2-meter temperature (T2M) predictions with distributional neural networks. For this, we use weather station data (Demaeyer et al., 2023) and the ensemble model of Feik et al. (2024); details are provided in Appendix C.

While the original task loss is the CRPS, we also train and evaluate the ensemble under alternative losses corresponding to the introduced scoring rules.

Figure 3 shows test instances sorted by decreasing total uncertainty, separately for each task loss and uncertainty measure. The figure reveals large differences across task losses, yet relatively minor variation between individual measures on a fixed task. For example, when training with squared error, none of the measures performs well: uncertain predictions do not translate into high loss, likely because squared error is not strictly proper. Interestingly, although unsuitable as a task loss, the uncertainty measure induced by $S_{SE}$ still behaves similarly to measures originating from strictly proper rules. This suggests that even when a scoring rule is not a good loss, its associated uncertainty measure may remain useful in practice. Further analyses of AU and EU are reported in Appendix C.

## 6.4 ACTIVE LEARNING

Here, we consider active learning as a downstream task, a standard benchmark for uncertainty measures. Here, the objective is to select new training instances under a budget, using epistemic uncertainty as the selection criterion (Hofman et al., 2024a; Nguyen et al., 2022; Kirsch et al., 2019). We estimate epistemic uncertainty with the pairwise estimator and the score divergence $D$ derived from each corresponding scoring rule. Here, we consider again the UCI benchmark dataset, of which we use five datasets and the T2M post-processing task. For the latter, we only use the ensemble approach, as used by the authors (Feik et al., 2024). For the UCI task, we utilize the deep ensemble Lakshminarayanan et al. (2017), deep evidential regression Amini et al. (2020), the CRPS ensemble (CRPS-ENS) by Kelen et al. (2025) and a mixture density network (MDN) (Bishop, 1994; Kelen et al., 2025). For details, compare Appendix C.

Table 3: Average rank (↓) of the different uncertainty measures and uncertainty representation methods averaged across all experiments. The best rank is highlighted in bold. $S_{\log}$ is not available for the sample-based methods.

| Experiment | | $S_{\log}$ | $S_{SE}$ | $S_{ES}$ | $S_{k_\gamma}$ |
|---|---|---|---|---|---|
| **UCI** | Deep Ensemble | 2.8 | **1.6** | 2.2 | 3.4 |
| | DER | **1.8** | 2.4 | 3.0 | 2.8 |
| | CRPS-ENS | - | 2.4 | 2.2 | **1.4** |
| | MDN | - | **1.6** | 2.4 | 2.0 |
| **T2M** | Deep Ensemble | 2.0 | 4.0 | 3.0 | **1.0** |
| Average | | 2.2 | 2.4 | 2.6 | **2.1** |

Figure 4 shows the performance of the different uncertainty measures in the T2M task. It is visible that the $S_{SE}$ (variance-based) measure performs significantly worse, while the other measures are indistinguishable from a performance perspective. One reasonable explanation is that in the more complex surface temperature task, the higher-order moments of the predictive distributions are highly relevant for assessing uncertainty. However, as shown in the previous section, the $S_{SE}$ measure is the only one that violates Proposition 5.1, due to it not being strictly proper.

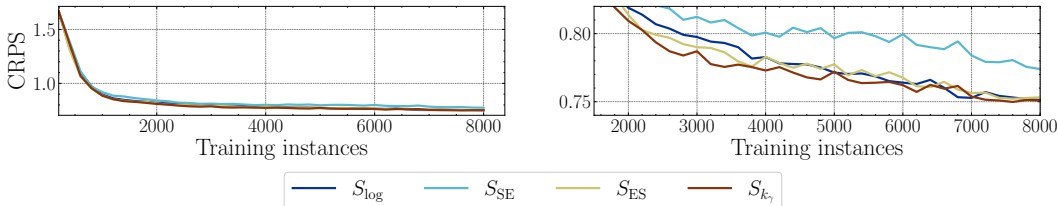

Figure 4: Continuous ranked probability score with increasing training instances for different model runs with the different uncertainty measures, averaged across three runs. The left panel shows the full run, the right panel shows a close-up after the first 2000 instances.

## 7 RELATED WORK

*Novel uncertainty measures.* Many studies focus on quantifying uncertainty for predictive models, especially for classification. While the most commonly used measures are based on the Shannon entropy (Houlsby et al., 2011), those have been criticized for having undesirable properties (Wimmer et al., 2023). Several generalizations have been proposed, such as variance-based Sale et al. (2023b), distance-based (Berry & Meger, 2024; Sale et al., 2023a) or pairwise (Schweighofer et al., 2023; Malinin & Gales, 2018; Berry & Meger, 2024) estimators. Closest to our work are recent developments in deriving uncertainty measures based on proper scoring rules and divergences. Gruber & Buettner (2023); Adlam et al. (2022) derive a bias-variance decomposition based on Bregman divergences that can be used for uncertainty quantification. Similarly, Gruber & Buettner (2024) derive a bias-variance decomposition specifically for kernel scores, where the corresponding uncertainty measures are conceptually similar to our proposed measures. However, their study focuses on generative models and on assessing predictive performance. Recently, (Kotelevskii et al., 2025; Hofman et al., 2024a;b; Schweighofer et al., 2023) introduced a framework for decomposing and quantifying uncertainty based on proper scoring rules and corresponding Bregman divergences. While similar in nature, our work specifically considers scoring rule-based uncertainty measures in the regression setting, which fundamentally differs from classification.

*Uncertainty quantification in regression.* While many works focus on uncertainty representation in regression, for example, via second-order distributions (Amini et al., 2020; Meinert & Lavin, 2022; Malinin et al., 2020) or ensembles (Berry & Meger, 2023; Lakshminarayanan et al., 2017; Kelen et al., 2025), little is usually done in the direction of analyzing the underlying uncertainty measures. The studies usually employ either the variance-based measure (Amini et al., 2020; Meinert & Lavin, 2022; Valdenegro-Toro & Mori, 2022) or (a variant of) the entropy-based measure (Malinin et al., 2020; Berry & Meger, 2024; Postels et al., 2021). While Bülte et al. (2025b) compare both measures with respect to a given set of preferable properties, they do not consider other measures or the pairwise variants thereof. In contrast, our work proposes a general framework to construct uncertainty measures in regression that can be used to derive many different instantiations of the measures with potentially different properties.

## 8 DISCUSSION

We propose a new framework for uncertainty quantification in supervised regression, based on strictly proper scoring rules and kernel scores. This framework generalizes recent advances from the classification setting, encompassing widely used uncertainty measures while also enabling the systematic construction of new ones. Our analysis highlights how specific properties of kernel scores directly translate into distinct characteristics of the induced uncertainty measures, offering a way to select measures based on underlying task requirements. Beyond the theoretical results, our numerical experiments demonstrate the versatility of the proposed measures, illustrating both their robustness and their adaptiveness to task-specific requirements.

**Limitations and future work** While our construction provides a principled foundation, it is not unique—alternative measures may satisfy the same properties. This opens up opportunities to develop criteria or selection procedures that help identify which measure is most appropriate in practice. Similarly, we focused on a specific set of properties, but many other aspects—such as efficiency or interpretability could enrich the framework and extend its applicability. On the empirical side, our study was primarily comparative within the proposed framework; extending evaluations to a wider spectrum of uncertainty quantification and uncertainty representation methods would offer deeper insights into its practical utility. Exciting opportunities also lie in exploring richer data domains, such as spatial, graph-structured, or functional data, where the interaction between kernel scores and domain structure could reveal new insights. Similarly, adapting the proposed measures to generalized kernel scores, such as weighted scores (Allen et al., 2023), could allow further tailoring of the measures for a specific task, such as the identification of extreme events. Finally, additional theoretical work on the relationship between kernel scores and maximum mean discrepancy may uncover additional properties and guide the principled design of task-specific "optimal" measures.

## Reproducibility statement

To ensure reproducibility, we only use publicly available datasets and model implementations. For datasets, we use the UCI benchmark (Hernández-Lobato & Adams, 2015), the WeatherBench2 benchmark (Rasp et al., 2024) and the EUPPBench benchmark (Demaeyer et al., 2023). In addition, we use the following model implementations: Distributional regression network (Rasp & Lerch, 2018; Feik et al., 2024), deep evidential regression (Amini et al., 2020) and implementations from the publicly available repository lightning-uq-box (Lehmann et al., 2025). Our own adaptations, implementations and reproducible experiments are available in an anonymous repository (`https://anonymous.4open.science/r/ke_anonymous-A80D`).

## Use of Large Language Models

Large language models (OpenAI's ChatGPT) were used to assist with improving grammar, style, and phrasing in the final stage of this manuscript.

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

# A   PROOFS

## A.1   PROOFS OF PROPOSITIONS 5.1 - 5.3

*Proof of Proposition 5.1.* Here we prove that for any proper scoring rule $S$, it holds that

1. $Q = \delta_{\mathbb{P}} \implies \mathrm{EU}(Q) = 0$, while for a *strictly* proper scoring rule the converse holds as well,

2. $\mathrm{EU}(\delta_{\mathbb{P}}) \leq \mathrm{EU}(Q_1) \leq \mathrm{EU}(Q_2)$.

1. Consider the BMA estimator. For $Q = \delta_{\mathbb{P}}$ we have $\overline{\mathbb{P}} = \mathbb{P}$ and $\mathrm{EU}(Q) = \mathbb{E}_{\mathbb{P} \sim Q}[D(\overline{\mathbb{P}}, \mathbb{P})] = D(\mathbb{P}, \mathbb{P}) = 0$, since $D$ is a divergence. For a strictly proper scoring rule, we obtain

$$\mathrm{EU}(Q) = \mathbb{E}_{\mathbb{P} \sim Q}[D(\overline{\mathbb{P}}, \mathbb{P})] = 0 \implies \overline{\mathbb{P}} = \mathbb{E}_Q[\mathbb{P}] = \mathbb{P} \implies Q = \delta_{\mathbb{P}}.$$

For the pairwise estimator, the proof works in an analogous way.

2 (BMA). The lower bound follows immediately from the nonnegativity of the divergence $D$ and the first part of the proposition being fulfilled for a proper scoring rule. Furthermore, we are given $Q_1 \leq_{\mathrm{cx}}^2 Q_2$ and $\mathrm{EU}(Q) = \mathbb{E}_{\mathbb{P} \sim Q}[D(\overline{\mathbb{P}}, \mathbb{P})]$. Recall that for a scoring rule with $\mathbb{P}, \mathbb{Q} \in \mathcal{P}(\mathcal{Y})$, the divergence is given as $D(\mathbb{P}, \mathbb{Q}) = S(\mathbb{P}, \mathbb{Q}) - S(\mathbb{Q}, \mathbb{Q})$. We want to show that

$$\mathrm{EU}(Q_1) = \mathbb{E}_{\mathbb{P} \sim Q_1}[D(\overline{\mathbb{P}}, \mathbb{P})] \leq \mathbb{E}_{\mathbb{P} \sim Q_2}[D(\overline{\mathbb{P}}, \mathbb{P})] = \mathrm{EU}(Q_2).$$

We will show that $D(\overline{\mathbb{P}}, \mathbb{P})$ is a convex functional in $\mathbb{P}$. Then, by definition of the convex order, it follows that $\mathrm{EU}(Q_1) \leq \mathrm{EU}(Q_2)$.

First, note that by definition of the convex order we have a fixed $\overline{\mathbb{P}} = \mathbb{E}_{\mathbb{P} \sim Q_1}[\mathbb{P}] = \mathbb{E}_{\mathbb{P} \sim Q_2}[\mathbb{P}]$. By definition of proper scoring rules, the term $S(\mathbb{P}, \mathbb{Q})$ is affine in $\mathbb{Q}$ (Dawid, 2007) and therefore convex. Furthermore, we know that $H(\mathbb{Q}) = S(\mathbb{Q}, \mathbb{Q})$ is a concave function in $\mathbb{Q}$ (Waghmare & Ziegel, 2025) and therefore $-H(\mathbb{Q})$ is convex. In total, $D(\overline{\mathbb{P}}, \mathbb{P})$ consists of an affine function plus a convex function in $\mathbb{P}$ and is therefore also convex in $\mathbb{P}$ (Boyd & Vandenberghe, 2004).

2 (Pairwise). For the pairwise estimator, we require the additional assumption that for a fixed $\mathbb{Q}$, the map $\mathbb{P} \mapsto S(\mathbb{P}, \mathbb{Q})$ is convex, which is fulfilled by kernel scores or scoring rules of Bregman type. First, write $F(\mathbb{P}, \mathbb{P}') := D(\mathbb{P} \,\|\, \mathbb{P}') = S(\mathbb{P}, \mathbb{P}') - H(\mathbb{P}')$.

By the convexity assumption, for fixed $\mathbb{P}'$, the map $\mathbb{P} \mapsto F(\mathbb{P}, \mathbb{P}')$ is convex. Furthermore, since $S(\mathbb{P}, \mathbb{Q})$ is affine in $\mathbb{Q}$ and $H(\mathbb{P})$ is concave (Dawid, 2007), for fixed $\mathbb{P}$, the map $\mathbb{P}' \mapsto F(\mathbb{P}, \mathbb{P}')$ is affine + convex, hence convex.

For every fixed $\mathbb{P}'$, we obtain the following via the convex order

$$\mathbb{E}_{\mathbb{P} \sim Q_1} F(\mathbb{P}, \mathbb{P}') \ \leq \ \mathbb{E}_{\mathbb{P} \sim Q_2} F(\mathbb{P}, \mathbb{P}').$$

Integrating over $\mathbb{P}' \sim Q_1$ gives

$$\mathbb{E}_{\mathbb{P}, \mathbb{P}' \sim Q_1} F(\mathbb{P}, \mathbb{P}') \ \leq \ \mathbb{E}_{\mathbb{P}' \sim Q_1} \mathbb{E}_{\mathbb{P} \sim Q_2} F(\mathbb{P}, \mathbb{P}').$$

Similarly, for every fixed $\mathbb{P}$, we obtain

$$\mathbb{E}_{\mathbb{P}' \sim Q_1} F(\mathbb{P}, \mathbb{P}') \ \leq \ \mathbb{E}_{\mathbb{P}' \sim Q_2} F(\mathbb{P}, \mathbb{P}').$$

Integrating over $\mathbb{P} \sim Q_2$ gives

$$\mathbb{E}_{\mathbb{P} \sim Q_2} \mathbb{E}_{\mathbb{P}' \sim Q_1} F(\mathbb{P}, \mathbb{P}') \ \leq \ \mathbb{E}_{\mathbb{P}, \mathbb{P}' \sim Q_2} F(\mathbb{P}, \mathbb{P}').$$

Since both sides coincide (by Fubini's theorem), we ultimately get

$$\mathbb{E}_{\mathbb{P}, \mathbb{P}' \sim Q_1} F(\mathbb{P}, \mathbb{P}') \ \leq \ \mathbb{E}_{\mathbb{P}, \mathbb{P}' \sim Q_2} F(\mathbb{P}, \mathbb{P}'),$$

i.e.

$$\mathrm{EU}_P(Q_1) \ \leq \ \mathrm{EU}_P(Q_2).$$

$\square$

*Proof of Proposition 5.2.* Here we prove that any kernel score $S_k$ with a translation invariant kernel $k(x, x')$ that is convex in one of its arguments fulfills $\mathrm{AU}(\delta_{\mathbb{P}_1}) \leq \mathrm{AU}(\delta_{\mathbb{P}_2})$.

We know by assumption that $\mathbb{P}_1 \leq_{\mathrm{cx}} \mathbb{P}_2$ and $\mathrm{AU}(\delta_{\mathbb{P}}) = H(\mathbb{P})$. Therefore, we need to show that $H(\mathbb{P}_1) \leq H(\mathbb{P}_2)$. Recall that for any translation invariant kernel score we have $k(x, x') = \psi(x - x')$ for some $\psi : \mathcal{Y} \to \mathbb{R}$ and the corresponding entropy is given as

$$H(\mathbb{P}) = \frac{1}{2}\mathbb{E}_{X,X'\sim\mathbb{P}}[k(X - X')] - \frac{1}{2}\mathbb{E}_{X\sim\mathbb{P}}[\underbrace{k(X - X)}_{\equiv k(0)}],$$

where the last part is a constant, due to the translation invariance, and therefore does not affect the inequality. Now define $\phi_P(x) := \mathbb{E}_{X'\sim\mathbb{P}}[\psi(x - X')]$, which is convex in x, since $\psi$ is convex and linearity in expectation preserves convexity.

Now, using convex order, we have

$$\mathbb{E}_{X\sim\mathbb{P}_1}[\phi_{\mathbb{P}_1}(X)] \leq \mathbb{E}_{Y\sim\mathbb{P}_2}[\phi_{\mathbb{P}_1}(Y)].$$

Similarly, we can also obtain an order for the convex function $\phi_{\mathbb{P}_2}$ as

$$\mathbb{E}_{X\sim\mathbb{P}_1}[\phi_{\mathbb{P}_2}(X)] \leq \mathbb{E}_{Y\sim\mathbb{P}_2}[\phi_{\mathbb{P}_2}(Y)].$$

Now, note that using Fubini's theorem, we obtain

$$\mathbb{E}_{X\sim\mathbb{P}_1}[\phi_{\mathbb{P}_2}(X)] = \mathbb{E}_{Y\sim\mathbb{P}_2}[\phi_{\mathbb{P}_1}(Y)] = \mathbb{E}_{U\sim\mathbb{P}_1, V\sim\mathbb{P}_2}[\psi(U - V)].$$

Therefore, we obtain

$$\mathbb{E}_{X\sim\mathbb{P}_1}[\phi_{\mathbb{P}_1}(X)] \leq \mathbb{E}_{Y\sim\mathbb{P}_2}[\phi_{\mathbb{P}_2}(Y)],$$

and therefore

$$\mathrm{AU}(\delta_{\mathbb{P}_1}) = H(\mathbb{P}_1) \leq H(\mathbb{P}_2) = \mathrm{AU}(\delta_{\mathbb{P}_2}).$$

$\square$

*Proof of Proposition 5.3.* We show the following: Consider a parametric first-order distributions $\mathbb{P}_{\boldsymbol{\theta}} \in \mathcal{P}(\mathcal{Y})$ with $\boldsymbol{\theta} \in \Theta \subseteq \mathbb{R}^p$, a corresponding second-order distributions $Q \in \mathcal{P}(\Theta)$, first-order distribution $\boldsymbol{\vartheta} \sim Q$ and assume that $\mathrm{AU}(Q) < \infty$. Furthermore, define $Q_\varepsilon := (1-\varepsilon)Q + \varepsilon\delta_{\boldsymbol{\theta}_0}$, $\boldsymbol{\theta}_0 \in \Theta$ and consider the influence function (IF):

$$\mathrm{IF}(\boldsymbol{\theta}_0; \mathrm{AU}, Q) = \lim_{\varepsilon\to 0} \frac{\mathrm{AU}(Q_\varepsilon) - \mathrm{AU}(Q)}{\varepsilon} = H(\mathbb{P}_{\boldsymbol{\theta}_0}) - \mathbb{E}_Q[H(\mathbb{P}_{\boldsymbol{\vartheta}})].$$

We then have that any kernel score $S_k$ with bounded kernel $k$ is robust in terms of the influence function.

Recall that $H_k(P_{\boldsymbol{\theta}_0}) = \frac{1}{2}\mathbb{E}_{X,X'\sim\mathbb{P}_{\boldsymbol{\theta}_0}}[k(X, X')] - \frac{1}{2}\mathbb{E}_{X\sim\mathbb{P}_{\boldsymbol{\theta}_0}}[k(X, X)]$. In particular, if $k$ is bounded, i.e. $k \leq C < \infty$ for some $C \in \mathbb{R}$ it follows from the linearity of expectation that $H_k(\mathbb{P}_{\boldsymbol{\theta}_0}) \leq C$ and therefore, with $\mathrm{AU}(Q) < \infty$ that $\mathrm{IF}(\boldsymbol{\theta}_0; \mathrm{AU}, Q) \leq C < \infty$. $\square$

## A.2 ADDITIONAL PROPOSITIONS FOR EXISTING MEASURES

Here, we introduce and prove two more propositions regarding the variance- and entropy-based measures.

**Proposition A.1.** *The variance-based measure (squared error) does not fulfill point 1 of Proposition 5.1.*

*Proof.* Consider the BMA estimator, two first-order Gaussian distribution, e.g. $\mathbb{P}_1 = \mathcal{N}(0, \sigma_1^2), \mathbb{P}_2 = \mathcal{N}(0, \sigma_2^2)$ with $\sigma_1^2 \neq \sigma_2^2$ and a second-order distribution, specified as a Dirac mixture, i.e. $Q = \frac{1}{2}\delta_{\mathbb{P}_1} + \frac{1}{2}\delta_{\mathbb{P}_2}$. Recall that for the variance-based measure, we have $D(\mathbb{P}, \mathbb{Q}) = (\mathbb{E}_{Y\sim\mathbb{P}}[Y] - \mathbb{E}_{Y'\sim\mathbb{Q}}[Y'])^2$. In addition, we obtain $\overline{\mathbb{P}} = \frac{1}{2}\mathbb{P}_1 + \frac{1}{2}\mathbb{P}_2$ and $\mathbb{E}_{Y'\sim\overline{P}}[Y'] = 0$. Then we obtain

$$\mathrm{EU}(Q) = \mathbb{E}_{\mathbb{P}\sim Q}[D(\overline{\mathbb{P}}, \mathbb{P})] = \mathbb{E}_{\mathbb{P}\sim Q}[(\underbrace{\mathbb{E}_{Y'\sim\overline{\mathbb{P}}}[Y']}_{=0} - \mathbb{E}_{Y\sim\mathbb{P}}[Y])^2] = \mathbb{E}_{\mathbb{P}\sim Q}[(\mathbb{E}_{Y\sim\mathbb{P}}[Y])^2]$$

$$= \frac{1}{2}\mathbb{E}_{\mathbb{P}_1\sim Q}[(\underbrace{\mathbb{E}_{Y\sim\mathbb{P}_1}[Y]}_{=0})^2] + \frac{1}{2}\mathbb{E}_{\mathbb{P}_2\sim Q}[(\underbrace{\mathbb{E}_{Y\sim\mathbb{P}_2}[Y]}_{=0})^2] = 0.$$

Therefore, we obtain $\mathrm{EU}(Q) = 0$ although $Q \neq \delta_{\mathbb{P}}$. The same argument also works for the pairwise estimator. $\qquad\square$

**Proposition A.2.** *The entropy-based measure (log-score) fulfills* $\mathrm{AU}(\delta_{\mathbb{P}_1}) \leq \mathrm{AU}(\delta_{\mathbb{P}_2})$ *if the underlying density is log-concave.*

*Proof.* A probability distribution has log-concave density, if the density can be expressed as $p(x) \equiv \exp(\varphi(x))$ for a concave function $\varphi(x)$. Recall that the log-score corresponds to the differential entropy, which can be expressed as

$$H(\mathbb{P}) = -\int p(x) \log p(x) d\mu(x) = \mathbb{E}_{\mathbb{P}}[-\log p(X)].$$

Then, for a log-concave density, we have that $\phi(x) := -\log p_2(x)$ is a convex function in $x$. By convex order, we then have

$$\mathbb{E}_{X \sim \mathbb{P}_1}[-\log p_2(X)] = \mathbb{E}_{X \sim \mathbb{P}_1}[\phi(X)] \leq \mathbb{E}_{Y \sim \mathbb{P}_2}[\phi(Y)] = H(\mathbb{P}_2).$$

The left-hand side is the cross-entropy of $\mathbb{P}_1, \mathbb{P}_2$, which, by definition, can be decomposed into

$$\mathbb{E}_{X \sim \mathbb{P}_1}[-\log p_2(X)] = H(\mathbb{P}_1) + D_{\mathrm{KL}}(\mathbb{P}_1 \| \mathbb{P}_2) \geq H(\mathbb{P}_1),$$

where the inequality follows from the KL-divergence being nonnegative. Combining the above gives

$$H(\mathbb{P}_1) \leq \mathbb{E}_{X \sim \mathbb{P}_1}[-\log p_2(X)] = \mathbb{E}_{X \sim \mathbb{P}_1}[\phi(X)] \leq \mathbb{E}_{Y \sim \mathbb{P}_2}[\phi(Y)] = H(\mathbb{P}_2),$$

and therefore

$$\mathrm{AU}(\delta_{\mathbb{P}_1}) = H(\mathbb{P}_1) \leq H(\mathbb{P}_2) = \mathrm{AU}(\delta_{\mathbb{P}_2}).$$

$\qquad\square$

# B  DERIVATION OF MEASURES FOR SPECIFIC CHOICES OF SCORING RULES

In this section, we derive expressions for the (generalized) entropy- and divergence term of the uncertainty measures introduced in this article. Recall that in order to assess EU, AU and TU, one requires expressions for the entropy, divergence and expected scoring rule. This is regardless whether one chooses the pairwise or the BMA estimator. Therefore, for $\mathbb{P}, \mathbb{Q} \in \mathcal{P}$ and $X, X' \sim \mathbb{P}$, $Y, Y' \sim \mathbb{Q}$, and $\mathbb{P}, \mathbb{P}' \sim Q$, $\overline{\mathbb{P}} = \mathbb{E}_Q[\mathbb{P}]$, we will derive the quantities $H(\mathbb{P}), D(\mathbb{P}, \mathbb{Q})$, as well as the gap between the BMA and pairwise estimation $\Delta$, for different scoring rules.

**Log-score** Let $\mathcal{P}$ be the set of distributions on $\mathcal{Y}$ that are absolutely continuous with respect to the Lebesgue measure $\mu$ and $\mathbb{P}, \mathbb{Q} \in \mathcal{P}$ with corresponding densities $p, q$. The *logarithmic score* $S_{\log} : \mathcal{P} \times \mathcal{Y} \to \overline{\mathbb{R}}$, given by

$$S_{\log}(\mathbb{P}, \boldsymbol{y}) = -\log p(\boldsymbol{y})$$

is a strictly proper scoring rule. The associated entropy and divergence are given as

$$H_{\log}(\mathbb{P}) = -\int p(\boldsymbol{x}) \log p(\boldsymbol{x}) \, d\mu(\boldsymbol{x}),$$

$$D_{\log}(\mathbb{P}, \mathbb{Q}) = \int q(\boldsymbol{y}) \log \left( \frac{q(\boldsymbol{y})}{p(\boldsymbol{y})} \right) d\mu(\boldsymbol{y}) = D_{\mathrm{KL}}(\mathbb{Q} \| \mathbb{P}),$$

which are the Shannon entropy and Kullback-Leibler divergence, respectively. Utilizing the BMA estimator, we obtain the entropy-based measure, while for the pairwise estimator we obtain the pairwise KL-divergence, as shown by Schweighofer et al. (2023). For their difference, we obtain the so-called reverse mutual information

$$\Delta = \mathbb{E}_Q \left[ D_{\mathrm{KL}} \left( \overline{\mathbb{P}} \| \mathbb{P} \right) \right].$$

**Kernel score** Consider the kernel score $S_k : \mathcal{P}_k \times \mathcal{Y}$ associated with a negative definite kernel $k$. We obtain the following expressions for the pairwise estimator:

$$H(\mathbb{P}) = \frac{1}{2} \mathbb{E}_{\mathbb{P}} \left[ k(X, X') \right] - \frac{1}{2} \mathbb{E}_{\mathbb{P}}[k(X, X)],$$

$$D(\mathbb{P}, \mathbb{Q}) = \mathbb{E}_{\mathbb{P}, \mathbb{Q}} \left[ k(X, Y) \right] - \frac{1}{2} \mathbb{E}_{\mathbb{P}} \left[ k(X, X') \right] - \frac{1}{2} \mathbb{E}_{\mathbb{Q}} \left[ K(Y, Y') \right].$$

The corresponding uncertainty measures are obtained by plugging the selected kernel into the above quantities.

**Squared error**    Let $\mathcal{P}$ be the set of distributions on $\mathcal{Y} \subseteq \mathbb{R}^p$ such that $\int \|x\|^2 \, d\mathbb{P}(x) < \infty$ and $Y \sim \mathbb{P} \in \mathcal{P}$. The squared error $S_{\mathrm{SE}} : \mathcal{P} \times \mathcal{Y} \to \overline{\mathbb{R}}$ given by

$$S_{\mathrm{SE}}(\mathbb{P}, y) = (y - \mathbb{E}_{\mathbb{P}}[Y])^2,$$

is a proper (but not strictly proper) kernel rule, with $k(x, x') = \|x - x'\|^2$. The associated entropy and divergence are given as

$$H_{\mathrm{SE}}(\mathbb{P}) = \mathrm{tr}(\mathrm{Cov}_{\mathbb{P}}[Y]), \qquad D_{\mathrm{SE}}(\mathbb{P}, \mathbb{Q}) = \|\boldsymbol{\mu}_{\mathbb{P}} - \boldsymbol{\mu}_{\mathbb{Q}}\|^2 \, .$$

In the case of the squared error, the corresponding uncertainty measures can be expressed in terms of moments of moments of the first order distribution, leading to the following measures for the BMA estimator:

$$\mathrm{AU}_B(Q) = \mathbb{E}_Q \left[ \mathrm{tr}(\mathrm{Cov}_{\mathbb{P}}[Y]) \right],$$

$$\mathrm{EU}_B(Q) = \mathbb{E}_Q \left[ \|\boldsymbol{\mu}_{\mathbb{P}} - \boldsymbol{\mu}_{\mathbb{P}'}\|^2 \right] = \mathrm{tr} \left( \mathrm{Cov}_Q[\boldsymbol{\mu}_{\mathbb{P}}] \right),$$

$$\mathrm{TU}_B(Q) = \mathbb{E}_Q \left[ \|Y - \mathbb{E}_Q[\boldsymbol{\mu}_{\mathbb{P}}]\|^2 \right],$$

which reduces to the variance-based decomposition in the univariate case $\mathcal{Y} \subseteq \mathbb{R}$. For the pairwise estimator, we obtain

$$\mathrm{AU}_P(Q) = \mathbb{E}_Q \left[ \mathrm{tr}(\mathrm{Cov}_{\mathbb{P}}[Y]) \right],$$

$$\mathrm{EU}_P(Q) = 2\mathbb{E}_Q \left[ \|\boldsymbol{\mu}_{\mathbb{P}} - \boldsymbol{\mu}_{\mathbb{P}'}\|^2 \right] = 2\mathrm{tr} \left( \mathrm{Cov}_Q[\boldsymbol{\mu}_{\mathbb{P}}] \right),$$

$$\mathrm{TU}_P(Q) = \mathbb{E}_Q \left[ \|Y - \mathbb{E}_Q[\boldsymbol{\mu}_{\mathbb{P}}]\|^2 \right] + \mathrm{tr} \left( \mathrm{Cov}_Q[\boldsymbol{\mu}_{\mathbb{P}}] \right),$$

which shows that both estimators only differ by a factor of two for the epistemic uncertainty. The gap between both estimators is

$$\Delta = \mathrm{tr} \left( \mathrm{Cov}_Q[\boldsymbol{\mu}_{\mathbb{P}}] \right) = \mathbb{E}_Q[D_{\mathrm{SE}}(\overline{\mathbb{P}}, \mathbb{P})].$$

This quantity measures the expected (score-) divergence between the BMA against all possible models.

### B.1    CLOSED-FORM EXPRESSIONS FOR GAUSSIANS

Here, we derive closed-form expressions for the entropy and divergence term of different scoring rules for first-order Gaussian and mixture of Gaussian distributions. Recall that for kernel scores $S_k$ with a conditionally negative definite kernel $k$, the entropy and divergence of two probability measures $\mathbb{P}, \mathbb{Q} \in \mathcal{P}(\mathcal{Y})$ are given as

$$H_k(\mathbb{P}) = \frac{1}{2}\mathbb{E}_{X,X'\sim\mathbb{P}}[k(X, X')] - \frac{1}{2}\mathbb{E}_{X\sim\mathbb{P}}[k(X, X)] \tag{10}$$

$$D_k(\mathbb{P}, \mathbb{Q}) = \mathbb{E}_{X\sim\mathbb{P},Y\sim\mathbb{Q}}[k(X, Y)] - \frac{1}{2}\mathbb{E}_{X,X'\sim\mathbb{P}}[k(X, X')] - \frac{1}{2}\mathbb{E}_{Y,Y'\sim\mathbb{Q}}[k(Y, Y')]. \tag{11}$$

Consider two first-order Gaussian distributions $X \sim \mathbb{P} = \mathcal{N}(\mu, \sigma^2)$, $Y \sim \mathbb{Q} = \mathcal{N}(\nu, \tau^2)$. Then we obtain the following expressions:

**Log-score**

$$H(\mathbb{P}) = \frac{1}{2} \log(2\pi e \sigma^2), \tag{12}$$

$$D(\mathbb{P}, \mathbb{Q}) = \log \left( \frac{\tau}{\sigma} \right) + \frac{\sigma^2 + (\mu - \nu)^2}{2\tau^2} - \frac{1}{2}. \tag{13}$$

These expressions are obtained via well-known results from the differential entropy and KL-divergence for Gaussian distributions.

**Squared error**

$$H(\mathbb{P}) = \sigma^2, \tag{14}$$

$$D(\mathbb{P}, \mathbb{Q}) = (\mu - \nu)^2. \tag{15}$$

*Proof.* For the entropy, we obtain

$$H(\mathbb{P}) = \frac{1}{2}\mathbb{E}_{X,X'\sim\mathbb{P}}[(X - X')^2)] = \frac{1}{2}\left(\mathbb{E}_{\mathbb{P}}[X^2] - 2\mathbb{E}_{\mathbb{P}}[X]\mathbb{E}_{\mathbb{P}}[X'] + \mathbb{E}_{\mathbb{P}}[X'^2]\right) = \mathbb{V}_{\mathbb{P}}[X] = \sigma^2.$$

In addition, we have that $\mathbb{E}_{X\sim\mathbb{P},Y\sim\mathbb{Q}}[(X - Y)^2] = \mathbb{E}_{\mathbb{P}}[X^2] - 2\mathbb{E}_{\mathbb{P}}[X]\mathbb{E}_{\mathbb{Q}}[Y] + \mathbb{E}_{\mathbb{Q}}[Y^2]$ such that for the divergence we obtain

$$\begin{aligned}
D(\mathbb{P}, \mathbb{Q}) &= \mathbb{E}_{\mathbb{P}}[X^2] - 2\mathbb{E}_{\mathbb{P}}[X]\mathbb{E}_{\mathbb{Q}}[Y] + \mathbb{E}_{\mathbb{Q}}[Y^2] - \mathbb{V}_{\mathbb{P}}[X] - \mathbb{V}_{\mathbb{Q}}[Y] \\
&= \mathbb{E}_{\mathbb{P}}[X^2] - 2\mathbb{E}_{\mathbb{P}}[X]\mathbb{E}_{\mathbb{Q}}[Y] + \mathbb{E}_{\mathbb{Q}}[Y^2] - \mathbb{E}_{\mathbb{P}}[X^2] + \mathbb{E}_{\mathbb{P}}[X]^2 - \mathbb{E}_{\mathbb{Q}}[Y^2] + \mathbb{E}_{\mathbb{Q}}[Y]^2 \\
&= \mathbb{E}_{\mathbb{P}}[X]^2 - 2\mathbb{E}_{\mathbb{P}}[X]\mathbb{E}_{\mathbb{Q}}[Y] + \mathbb{E}_{\mathbb{Q}}[Y]^2 = (\mathbb{E}_{\mathbb{P}}[X] - \mathbb{E}_{\mathbb{Q}}[Y])^2 \\
&= (\mu - \nu)^2.
\end{aligned}$$

$\square$

**CRPS**

$$H(\mathbb{P}) = \frac{\sigma}{\sqrt{\pi}}, \tag{16}$$

$$D(\mathbb{P}, \mathbb{Q}) = \left(\sqrt{\sigma^2 + \tau^2}\right)\frac{\sqrt{2}}{\sqrt{\pi}}{}_1F_1\left(-\frac{1}{2}, \frac{1}{2}; -\frac{1}{2}\frac{(\mu-\nu)^2}{\sigma^2+\tau^2}\right) - \left(\frac{\sigma+\tau}{\sqrt{\pi}}\right). \tag{17}$$

*Proof.* Winkelbauer (2014) show that for the raw absolute moment of a Gaussian we have

$$\mathbb{E}[|X|^p] = \sigma^p 2^{p/2}\frac{\Gamma(\frac{p+1}{2})}{\sqrt{\pi}}{}_1F_1\left(-\frac{p}{2}, \frac{1}{2}; -\frac{\mu^2}{2\sigma^2}\right),$$

where ${}_1F_1$ denotes Kummer's confluent hypergeometric function. Furthermore, we know that $X - Y \sim \mathcal{N}(\mu - \nu, \sigma^2 + \tau^2)$, $X - X' \sim \mathcal{N}(0, 2\sigma^2)$ and $Y - Y' \sim \mathcal{N}(0, 2\tau^2)$. Therefore, we obtain

$$H(\mathbb{P}) = \frac{1}{2}\mathbb{E}_{X,X'\sim\mathbb{P}}[|X - X'|] = \frac{1}{2}\sqrt{2\sigma^2}\sqrt{2}\frac{\Gamma(1)}{\sqrt{\pi}}{}_1F_1\left(-\frac{1}{2}, \frac{1}{2}; 0\right) = \frac{\sigma}{\sqrt{\pi}}.$$

With $\mathbb{E}_{X\sim\mathbb{P},Y\sim\mathbb{Q}}[|X - Y|] = \sqrt{\sigma^2 + \tau^2}\frac{\sqrt{2}}{\sqrt{\pi}}{}_1F_1\left(-\frac{1}{2}, \frac{1}{2}; -\frac{1}{2}\frac{(\mu-\nu)^2}{\sigma^2+\tau^2}\right)$ we obtain the divergence $D(\mathbb{P}, \mathbb{Q})$ by plugging in the corresponding expectations. $\square$

**Gaussian kernel score**  Given the (negative) Gaussian kernel $k(x, y) = -\exp(-(x - y)^2/\gamma^2)$ with scalar bandwidth $\gamma$, we obtain

$$H(\mathbb{P}) = \frac{1}{2}\left(1 - \frac{\gamma}{\sqrt{\gamma^2 + 4\sigma^2}}\right) \tag{18}$$

$$D(\mathbb{P}, \mathbb{Q}) = \frac{1}{2}\frac{\gamma}{\sqrt{\gamma^2 + 4\sigma^2}} + \frac{1}{2}\frac{\gamma}{\sqrt{\gamma^2 + 4\tau^2}} - \frac{\gamma}{\sqrt{\gamma^2 + 2(\sigma^2 + \tau^2)}}\exp\left(-\frac{(\mu-\nu)^2}{\gamma^2 + 2(\sigma^2 + \tau^2)}\right) \tag{19}$$

*Proof.* Let $Z := X - Y \sim \mathbb{P}_Z := \mathcal{N}(\delta, \upsilon)$ with $\delta := \mu - \nu, \upsilon := \sigma^2 + \tau^2$. Then $\frac{Z^2}{\delta}$ follows a noncentral chi-squared distribution, i.e. $\frac{Z^2}{\delta} \sim \chi^2(1, \lambda)$ with noncentrality parameter $\lambda = \frac{\delta^2}{\upsilon}$. Furthermore, we have

$$\mathbb{E}_{X\sim\mathbb{P},Y\sim\mathbb{Q}}[k(X, Y)] = -\mathbb{E}_{\mathbb{P}_Z}\left[\exp\left(-\frac{\frac{Z^2}{\upsilon}\upsilon}{\gamma^2}\right)\right] = -M_{\chi^2(1,\lambda)}\left(-\frac{\upsilon}{\gamma^2}\right).$$

Here, $M_{\chi^2(k,\lambda)}(t)$ is the moment-generating function of $\chi^2(k,\lambda)$, with $t = -\frac{v}{\gamma^2}$, which can be expressed analytically (compare, for example, Patnaik (1949)) as $M_{\chi^2(k,\lambda)}(t) = \frac{\exp\left(\frac{\lambda t}{1-2t}\right)}{(1-2t)^{k/2}}$. Therefore, we obtain

$$\mathbb{E}_{X\sim\mathbb{P},Y\sim\mathbb{Q}}[k(X,Y)] = -\frac{\gamma}{\sqrt{\gamma^2 + 2(\sigma^2 + \tau^2)}} \exp\left(-\frac{(\mu - \nu)^2}{\gamma^2 + 2(\sigma^2 + \tau^2)}\right)$$

and

$$H(\mathbb{P}) = \frac{1}{2}\mathbb{E}_{X,X'\sim\mathbb{P}}[k(X,X')] - \frac{1}{2}\mathbb{E}_{X\sim\mathbb{P}}[k(X,X)]$$
$$= \frac{1}{2}\left(1 - \frac{\gamma}{\sqrt{\gamma^2 + 4\sigma^2}}\right).$$

By plugging these expressions into the definition of the divergence $D(\mathbb{P},\mathbb{Q})$, we obtain the corresponding closed form. □

**Gaussian mixtures** Here, we consider a mixture of Gaussians, i.e. $X \sim \mathbb{P} = \sum_{i=1}^{M} w_i \mathcal{N}(\mu_i, \sigma_i^2), Y \sim \mathbb{Q} = \sum_{j=1}^{N} v_j \mathcal{N}(\mu_j, \sigma_j^2)$ with nonnegative weights $w_i, v_j$ that sum to one. For a mixture of Gaussians, closed-form expressions are not necessarily available, as is the case for the log-score. However, for specific cases, closed-form expressions are available via the corresponding marginals. For a translation-invariant kernel score, the expressions for the mixture density network can be derived in terms of the kernel score of the individual components. By linearity of the expectation, we obtain

$$\mathbb{E}[k(X,Y)] = \sum_{i=1}^{M}\sum_{j=1}^{N} w_i v_j \mathbb{E}_{X\sim\mathcal{N}(\mu_i,\sigma_i^2),Y\sim\mathcal{N}(\mu_j,\sigma_j^2)}[k(X,Y)].$$

In the case of a translation invariant kernel, i.e. $k(X,Y) \equiv k(X-Y)$ this reduces to a weighted sum of the corresponding Gaussian score, as we have $X - Y \sim \mathcal{N}(\mu_i - \mu_j, \sigma_i^2 + \sigma_j^2)$. Therefore, we can use the results from the previous section to derive the scores for the Gaussian mixtures analytically.

**Marginal scores** In the multivariate setting $\mathcal{Y} \subseteq \mathbb{R}^d$ for $d > 1$, closed-form expressions are more difficult to obtain then in the univariate setting. For instance, for a Gaussian distribution, the energy score admits an analytic solution for $\beta = 1, d = 1$ but not for $\beta = 1, d > 1$. However, one can always define a multivariate strictly proper scoring rule from a univariate one. Let $\{Y_j\}_{j=1}^{d}$ be a collection of marginal distributions from the multivariate random variable $\mathbf{Y}$. Then one can construct a *marginal score* for $\mathbf{Y}$ as

$$S_M(\mathbb{P}, y) = \sum_{j=1}^{d} S(\mathbb{P}_j, y_j),$$

where $Y_j \sim \mathbb{P}_j$ when $\mathbf{Y} \sim \mathbb{P}$ and $S$ is a (strictly) proper scoring rule for the marginal $Y_j$. Then, the scoring rule $S_M$ is also strictly proper. This is especially interesting if the main interest is in the marginals, for example, if the dependence structure across the marginals is of little interest.

## C EXPERIMENT DETAILS

### C.1 OUT-OF-DISTRIBUTION DETECTION

**T2M** We follow the experiment setup in Bülte et al. (2025a) and use DRNs to post-process 2-meter surface temperature (T2M) predictions. More specifically, the input to the DRNs is the mean prediction of the ECMWF integrated (ensemble) forecast system (IFS), and the networks are trained to predict the parameters $\mu_\theta, \sigma_\theta^2$ of a Gaussian distribution per individual gridpoint. Similar to Bülte et al. (2025a), we use ERA5 data (Hersbach et al., 2020) with a spatial resolution of $0.25° \times 0.25°$ and a time resolution of $6h$. Furthermore, we restrict the data to a European domain, covering an area from 35°N – 75°N and 12.5°W – 42.5°E with selected user-relevant weather variables (u-component and v-component of 10-m wind speed (U10 and V10), temperature at 2m and 850 hPa

(T2M and T850), geopotential height at 500 hPa (Z500), as well as land-sea mask and orography) that serve as input to the model. In addition, we use a positional embedding of the latitude/longitude of each gridpoint, which improves model performance (Rasp & Lerch, 2018). All data is obtained via the WeatherBench2 repository (Rasp et al., 2024), a visualization of the domain, land-sea-mask and orography can be seen in Figure 5.

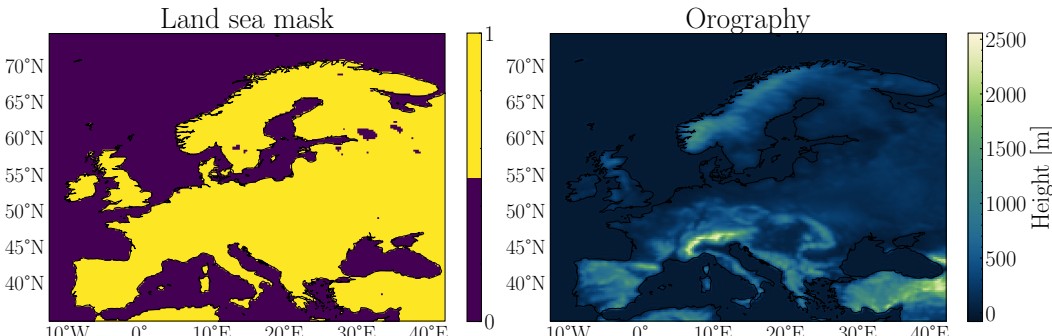

Figure 5: The figure shows the spatial domain used for the distributional regression networks, as well as the corresponding land-sea mask and orography.

We train an ensemble of $M = 10$ DRNs, with hyperparameters from Bülte et al. (2025a). During training, the models only see the land area of the domain, which allows to evaluate the uncertainty measures on out-of-distribution data.

To verify the results against a different uncertainty representation, we repeat the experiment using the deep evidential regression framework (Amini et al., 2020). In this setting, we have a first-order Gaussian and a second-order normal-inverse-gamma (NIG) distribution. We follow Amini et al. (2020) and use an additional regularization term for which we use different values $\lambda$. To obtain the uncertainty measures, we sample from the NIG distribution and use empirical (pairwise) estimates of TU, EU and AU, respectively. Figure 6 shows AU and EU for different values of $\lambda$. While the estimated uncertainties heavily depend on the regularization parameter, it is evident that the $S_{\text{SE}}$ are impacted by pointwise outliers, as the corresponding uncertainty values are very high. In contrast, the measures based on $S_{\text{ES}}$ and $S_{k_\gamma}$ seem to exhibit the structural changes across the topography of the domain most clearly.

**Depth estimation**   Similar to Amini et al. (2020), we use the NYU Depth v2 dataset (Silberman et al., 2012), which consists of image-depth pairs of indoor scenes. After training, the models are evaluated on ApolloScape (Huang et al., 2018), an OOD dataset of outdoor driving scenes. We utilize the same UNet backbone as (Amini et al., 2020) for each model and train ... . As uncertainty representation methods, we use the following:

1. Deep evidential regression, as implemented for this task in (Amini et al., 2020).

2. The multivariate CRPS model by Kelen et al. (2025) optimized using the energy score and with MCDropout (Gal & Ghahramani, 2016) to generate the second-order distribution.

3. A first-order predictive (multivariate) Gaussian, similar to Muschinski et al. (2024), with the covariance matrix parameterized using a low-rank + diagonal approximation (Rezende et al., 2014). The second-order distribution is again generated using MCDropout (Gal & Ghahramani, 2016).

## C.2   ROBUSTNESS

Here, we use a deep ensemble (Lakshminarayanan et al., 2017) on the concrete, energy and yacht dataset from the UCI regression benchmark (Hernández-Lobato & Adams, 2015). We train a base ensemble of $M = 25$ and $M = 5$ members and one additional member that is trained on a distorted target $\hat{y} = y + \mathcal{N}(0, \delta^2)$. This allows us to analyze the robustness of the different uncertainty measures with respect to an outlier in the ensemble prediction. Table 4 shows the mean absolute

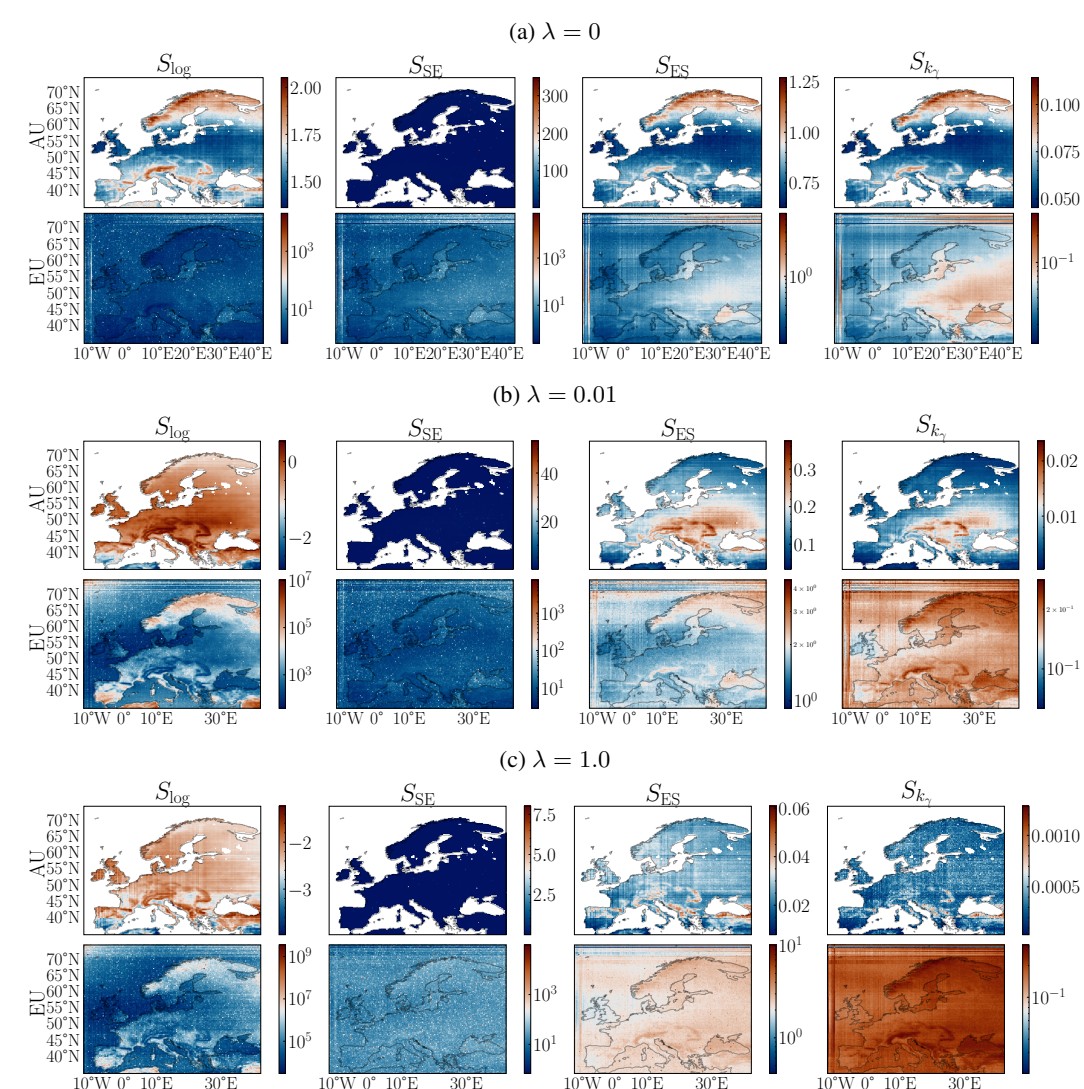

Figure 6: The figure shows AU and EU averaged over a test set of 365 days for the different uncertainty measures using deep evidential regression. For visualization purposes, epistemic uncertainty is shown on a log-scale.

percentage error of the aleatoric uncertainty from the base ensemble for different values of $\delta$ and different ensemble sizes. Table 5 shows the same for epistemic uncertainty. Figure 7 shows corresponding visualizations for the different datasets.

In addition to the results on the UCI benchmark, we can provide a theoretical analysis of the robustness in the case of a deep ensemble, which admits a first-order predictive Gaussian distribution $p(y \mid \boldsymbol{\theta}) = \mathcal{N}(\mu, \sigma^2), \boldsymbol{\theta} = (\mu, \sigma^2)^\top$. Assume that the second-order distribution fulfills $\|\mathbb{E}_Q[H(P_{\boldsymbol{\vartheta}})]\| < \infty$, meaning that the aleatoric uncertainty of the sample distribution $Q$ is well defined[3]. In that case, we can analyze the influence function $\mathrm{IF}(\boldsymbol{\theta}_0; \mathrm{AU}, Q)$ by analyzing the limit $\lim_{\boldsymbol{\theta}_0 \to \infty} H(P_{\boldsymbol{\theta}_0})$, since $\mathbb{E}_Q[H(P_{\boldsymbol{\vartheta}})]$ is a finite constant. Table 6 shows the closed-form expressions for $H(\boldsymbol{\theta}_0)$, as well as the corresponding growth rates in the contamination $\boldsymbol{\theta}_0$. While the Gaussian kernel score is the only scoring rule that is robust, since it admits a bounded influence function, the log-score and CRPS have a notably slower growth rate in $\boldsymbol{\theta}_0$ as the variance-based measure, which grows linearly with $\sigma_0^2$.

---

[3]For a finite ensemble this always holds.

Table 4: Effect of the added noise $\delta$ on the different aleatoric uncertainty measures for different ensemble sizes $M$ across all three datasets. The reported values are the mean absolute percentage error from the corresponding measure for the base ensemble.

| Experiment | $M$ | $S$ | 0.0 | 0.2 | 0.5 | 1.5 | 2.5 | 5.0 |
|---|---|---|---|---|---|---|---|---|
| **Concrete** | 5 | $S_{\log}$ | 1.20 | 4.15 | 6.76 | 14.5 | 19.3 | 20.8 |
| | | $S_{\mathrm{SE}}$ | 6.51 | 122 | 1.53e+03 | 3.49+e04 | 3.30e+05 | 2.10e+06 |
| | | $S_{\mathrm{ES}}$ | 3.06 | 22.2 | 66.7 | 356 | 1.04e+03 | 2.27e+03 |
| | | $S_{k_\gamma}$ | 0.14 | 0.47 | 0.60 | 0.80 | 0.84 | 0.88 |
| | 25 | $S_{\log}$ | 0.25 | 0.90 | 1.56 | 3.34 | 4.47 | 4.82 |
| | | $S_{\mathrm{SE}}$ | 1.11 | 25.3 | 324 | 7.21e+03 | 6.77e+04 | 4.78e+05 |
| | | $S_{\mathrm{ES}}$ | 0.55 | 4.7 | 14.7 | 78.2 | 224 | 503 |
| | | $S_{k_\gamma}$ | 0.03 | 0.10 | 0.13 | 0.18 | 0.19 | 0.19 |
| **Energy** | 5 | $S_{\log}$ | 0.52 | 2.45 | 5.07 | 9.86 | 11.4 | 16.4 |
| | | $S_{\mathrm{SE}}$ | 5.49 | 47.3 | 288 | 1.21e+04 | 2.93e+04 | 5.06e+07 |
| | | $S_{\mathrm{ES}}$ | 2.50 | 15.4 | 49.6 | 270 | 417 | 8.58e+03 |
| | | $S_{k_\gamma}$ | 1.64 | 6.27 | 10.2 | 13.1 | 13.5 | 14.0 |
| | 25 | log | 0.11 | 0.57 | 1.17 | 2.28 | 2.62 | 3.79 |
| | | $S_{\mathrm{SE}}$ | 1.12 | 10.6 | 64.1 | 2.86e+03 | 6.57e+03 | 1.09e+07 |
| | | $S_{\mathrm{ES}}$ | 0.52 | 3.52 | 11.3 | 62.5 | 95.3 | 1.933e+03 |
| | | $S_{k_\gamma}$ | 0.34 | 1.41 | 2.31 | 2.97 | 3.07 | 3.17 |
| **Yacht** | 5 | $S_{\log}$ | 0.30 | 5.03 | 8.97 | 11.9 | 15.6 | 18.6 |
| | | $S_{\mathrm{SE}}$ | 2.05 | 1.02e+03 | 1.37+e04 | 9.85+e05 | 2.83+e06 | 2.58+e07 |
| | | $S_{\mathrm{ES}}$ | 1.07 | 69.7 | 255 | 1.33e+03 | 3.07e+03 | 1.02e+04 |
| | | $S_{k_\gamma}$ | 0.63 | 11.7 | 14.1 | 14.8 | 15.5 | 15.6 |
| | 25 | $S_{\log}$ | 0.09 | 1.17 | 2.08 | 2.75 | 3.61 | 4.30 |
| | | $S_{\mathrm{SE}}$ | 0.60 | 225 | 2.88+e03 | 2.10+e05 | 6.42+e05 | 6.10+e06 |
| | | $S_{\mathrm{ES}}$ | 0.31 | 15.9 | 57.5 | 298 | 699 | 2.36e+03 |
| | | $S_{k_\gamma}$ | 0.18 | 2.65 | 3.18 | 3.36 | 3.52 | 3.54 |

## C.3    TASK ADAPTION

We use the distributional regression network from (Feik et al., 2024), which is used to post-process 2-meter surface temperature forecasts with a lead time of 24h on a station-based benchmark dataset (Demaeyer et al., 2023). The model issues a prediction at every individual station and is optimized and evaluated using the continuous ranked probability score. We use the hyperparameters from Feik et al. (2024). For analyzing the different measures, we train different ensembles ($M = 10$) with the different scoring rules as task losses and analyze the different measures of total uncertainty for each model. Figure 8 shows an additional visualization for the sorted epistemic and aleatoric uncertainty, respectively. While the behavior for AU looks similar to that of TU (compare Figure 3), for EU the measures behave very differently. For example, for all task losses except $S_{\mathrm{SE}}$, the measures $S_{\mathrm{SE}}$ and $S_{k_\gamma}$ show opposite behavior, i.e. one is decreasing, while the other is increasing. In these cases, epistemic uncertainty most likely does not contribute much to the total uncertainty and is therefore not aligned with the corresponding task loss. Instead, the task loss is highest whenever aleatoric uncertainty is highest.

Table 5: Effect of the added noise $\delta$ on the different epistemic uncertainty measures for different ensemble sizes $M$ across all three datasets. The reported values are the mean absolute percentage error from the corresponding measure for the base ensemble.

| Experiment | $M$ | $S$ | 0.0 | 0.2 | 0.5 | 1.5 | 2.5 | 5.0 |
|---|---|---|---|---|---|---|---|---|
| **Concrete** | 5 | $S_{\log}$ | 23.2 | 2.43e+03 | 1.31e+04 | 1.41e+05 | 2.9e+05 | 1.28e+06 |
| | | $S_{\mathrm{SE}}$ | 25.7 | 4.79e+03 | 1.82e+04 | 2.74e+05 | 3.38e+05 | 2.35e+06 |
| | | $S_{\mathrm{ES}}$ | 17.6 | 584 | 1.46e+03 | 5.2e+03 | 5.76e+03 | 1.97e+04 |
| | | $S_{k_\gamma}$ | 9.81 | 62.6 | 73 | 68.4 | 67.5 | 75.5 |
| | 25 | $S_{\log}$ | 3.49 | 499 | 3.1e+03 | 3.32e+04 | 5.29e+04 | 2.43e+05 |
| | | $S_{\mathrm{SE}}$ | 3.71 | 949 | 4.08e+03 | 6.17e+04 | 6.22e+04 | 4.74e+05 |
| | | $S_{\mathrm{ES}}$ | 2.68 | 116 | 335 | 1.09e+03 | 1.16e+03 | 4.05e+03 |
| | | $S_{k_\gamma}$ | 1.57 | 11.3 | 14.7 | 12.1 | 11.7 | 13.3 |
| **Energy** | 5 | $S_{\log}$ | 28.2 | 2.55e+03 | 2.78e+04 | 1.25e+05 | 4.34e+05 | 6.59e+06 |
| | | $S_{\mathrm{SE}}$ | 29.9 | 4.02e+03 | 5.17e+04 | 2.46e+05 | 8.58e+05 | 2.46e+06 |
| | | $S_{\mathrm{ES}}$ | 18.4 | 390 | 1.28e+03 | 3.23e+03 | 6.43e+03 | 1.02e+04 |
| | | $S_{k_\gamma}$ | 7.38 | 6.72 | 6.62 | 9.33 | 9.84 | 10.3 |
| | 25 | $S_{\log}$ | 3.33 | 386 | 2.94e+03 | 1.84e+04 | 7.74e+04 | 6.45e+05 |
| | | $S_{\mathrm{SE}}$ | 3.57 | 616 | 5.47e+03 | 3.62e+04 | 1.52e+05 | 3.12e+05 |
| | | $S_{\mathrm{ES}}$ | 2.38 | 74.9 | 226 | 605 | 1.26e+03 | 1.84e+03 |
| | | $S_{k_\gamma}$ | 0.878 | 1.12 | 1.19 | 1.98 | 2.08 | 2.24 |
| **Yacht** | 5 | $S_{\log}$ | 25.4 | 1.87e+05 | 3.55e+05 | 2.42e+06 | 2.14e+07 | 2.95e+07 |
| | | $S_{\mathrm{SE}}$ | 29.5 | 3.78e+05 | 7.2e+05 | 4.02e+06 | 3.68e+07 | 2.79e+07 |
| | | $S_{\mathrm{ES}}$ | 23.4 | 8.45e+03 | 1.28e+04 | 2.71e+04 | 8.19e+04 | 7.52e+04 |
| | | $S_{k_\gamma}$ | 16.7 | 68.8 | 70 | 67.6 | 64.4 | 64.4 |
| | 25 | $S_{\log}$ | 6.04 | 2.43e+04 | 4.39e+04 | 3.89e+05 | 2.34e+06 | 3.77e+06 |
| | | $S_{\mathrm{SE}}$ | 6.58 | 4.95e+04 | 8.69e+04 | 5.53e+05 | 4.06e+06 | 3.31e+06 |
| | | $S_{\mathrm{ES}}$ | 4.87 | 1.25e+03 | 1.71e+03 | 4.13e+03 | 1.12e+04 | 1.02e+04 |
| | | $S_{k_\gamma}$ | 2.75 | 10.3 | 9.13 | 8.49 | 8.32 | 8.31 |

Table 6: Limit and corresponding growth rates for the influence function $\mathrm{IF}(\boldsymbol{\theta}_0; \mathrm{AU}, Q)$ in the limit $\boldsymbol{\theta}_0 \to \infty$.

| $S$ | $H(P_{\boldsymbol{\theta}_0})$ | $\lim_{\boldsymbol{\theta}_0 \to \infty} H(P_{\boldsymbol{\theta}_0})$ | Growth |
|---|---|---|---|
| $S_{\log}$ | $\frac{1}{2}\log(2\pi e \sigma_0^2)$ | $\infty$ | $\mathcal{O}(\log(\sigma_0^2))$ |
| $S_{\mathrm{SE}}$ | $\sigma_0^2$ | $\infty$ | $\mathcal{O}(\sigma_0^2)$ |
| $S_{\mathrm{ES}}$ | $\frac{\sigma_0}{\sqrt{\pi}}$ | $\infty$ | $\mathcal{O}(\sqrt{\sigma_0^2})$ |
| $S_{k_\gamma}$ | $\frac{1}{2}\left(1 - \frac{\gamma}{\sqrt{\gamma^2 + 4\sigma_0^2}}\right)$ | 0.5 | $\mathcal{O}(1/\sqrt{\sigma_0^2})$ |

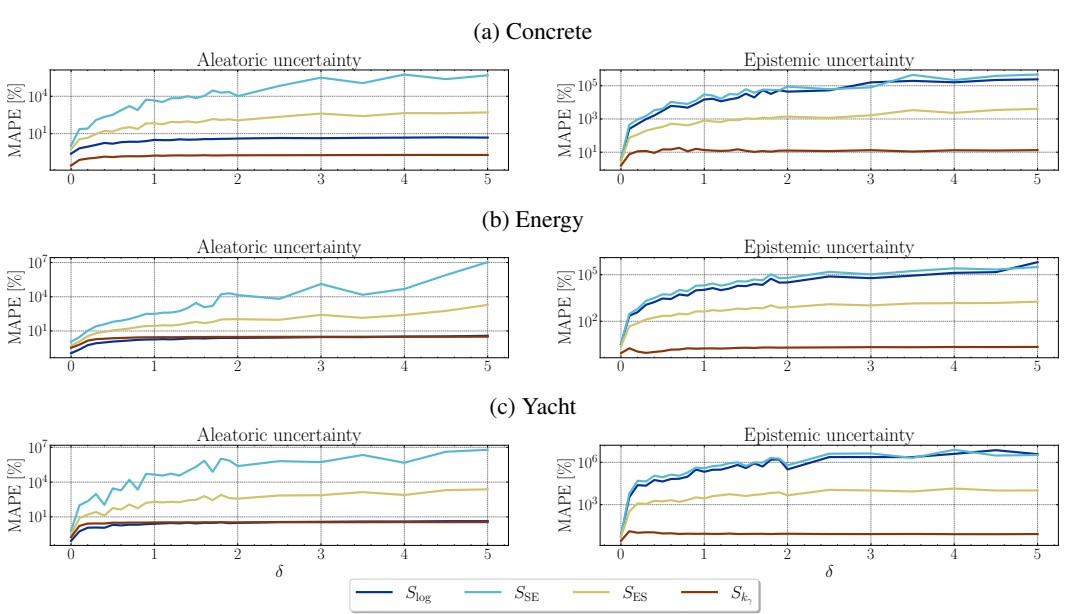

Figure 7: Effect of the added noise $\delta$ on the different (aleatoric) uncertainty measures for an ensemble of size $M = 25$ across all three datasets. The reported values are the mean absolute percentage error from the corresponding measure for the base ensemble.

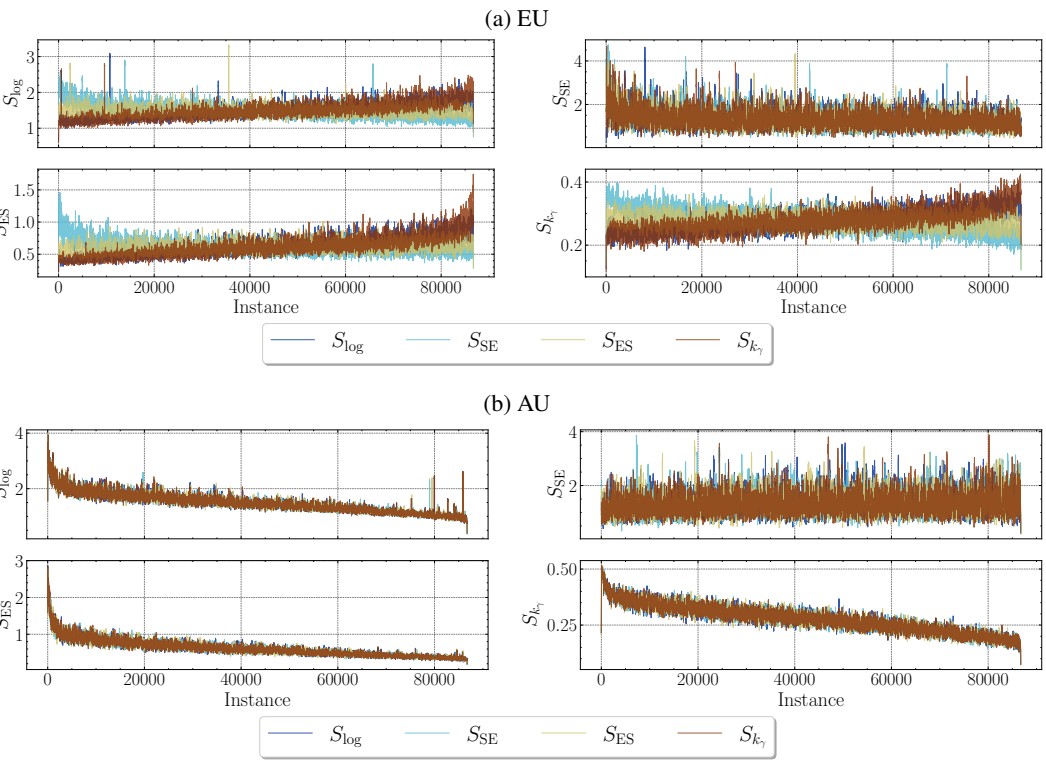

Figure 8: The figure shows the different task losses (each plot) sorted by each of the different uncertainty measures from highest to lowest epistemic (a) and aleatoric (b) uncertainty. For visualization purposes, the values shown are moving averages of size 50.

## C.4 ACTIVE LEARNING

Table 7: Performance of the different uncertainty measures for different uncertainty measures methods across the different experiments and uncertainty representation methods. The best model for each dataset and method is highlighted in bold and standard deviations are given in brackets.

| Experiment | Method | $S_{\log}$ | $S_{\text{SE}}$ | $S_{\text{ES}}$ | $S_{k_\gamma}$ |
|---|---|---|---|---|---|
| Energy | Deep Ensemble | 0.155 (0.004) | **0.133 (0.007)** | 0.142 (0.008) | 0.154 (0.002) |
| | DER | 0.137 (0.011) | **0.126 (0.007)** | 0.141 (0.010) | 0.156 (0.012) |
| | CRPS-ENS | — | 0.185 (0.006) | 0.183 (0.004) | **0.178 (0.008)** |
| | MDN | — | **0.186 (0.005)** | 0.190 (0.003) | 0.190 (0.008) |
| Concrete | Deep Ensemble | 0.267 (0.008) | **0.264 (0.009)** | 0.271 (0.006) | 0.274 (0.013) |
| | DER | 0.298 (0.017) | 0.313 (0.022) | 0.309 (0.030) | **0.290 (0.015)** |
| | CRPS-ENS | — | 0.341 (0.012) | 0.342 (0.009) | **0.340 (0.007)** |
| | MDN | — | 0.358 (0.019) | 0.358 (0.022) | **0.342 (0.009)** |
| Yacht | Deep Ensemble | 0.077 (0.005) | **0.051 (0.003)** | 0.058 (0.005) | 0.080 (0.004) |
| | DER | **0.060 (0.008)** | 0.061 (0.007) | 0.066 (0.012) | 0.092 (0.020) |
| | CRPS-ENS | — | 0.283 (0.005) | **0.282 (0.006)** | 0.287 (0.005) |
| | MDN | — | **0.242 (0.007)** | 0.243 (0.009) | 0.249 (0.007) |
| Boston | Deep Ensemble | 0.279 (0.010) | **0.258 (0.003)** | 0.264 (0.016) | 0.269 (0.012) |
| | DER | 0.299 (0.026) | 0.317 (0.017) | 0.302 (0.020) | **0.291 (0.009)** |
| | CRPS-ENS | — | 0.361 (0.009) | 0.360 (0.010) | **0.351 (0.004)** |
| | MDN | — | **0.337 (0.010)** | 0.339 (0.013) | 0.338 (0.006) |
| Naval | Deep Ensemble | **0.560 (0.048)** | 0.616 (0.075) | 0.563 (0.046) | 0.614 (0.069) |
| | DER | 0.370 (0.024) | **0.367 (0.037)** | 0.371 (0.074) | 0.547 (0.061) |
| | CRPS-ENS | — | 0.581 (0.016) | 0.588 (0.014) | **0.576 (0.006)** |
| | MDN | — | 0.574 (0.016) | 0.574 (0.012) | **0.565 (0.004)** |
| T2M | Deep Ensemble | 0.752 (0.011) | 0.774 (0.016) | 0.753 (0.005) | **0.751 (0.003)** |

**UCI**  Here, we use the datasets *energy, concrete, boston, naval*, and *yacht*. For the active learning task, we consider the following uncertainty representation methods:

The different models are trained for 3 epochs in each round, initialized with 20 samples and can acquire 10 new samples in each of 25 rounds. The results are averaged over five different runs. The model performance is evaluated on a held-out test set using the continuous ranked probability score.

**T2M**  For the active learning task, we train an ensemble of 10 DRNs that are initially trained using 200 samples and can acquire 200 new instances in each of 40 rounds. The models are trained for 2 epochs in each round. At the end of the 40 rounds, the model had access to around 10% of the training data. The model performance is evaluated using the continuous ranked probability score over the test set.

Table Table 7 shows the test loss after the active learning runs for the UCI and T2M datasets, as well as the different first-order prediction methods.

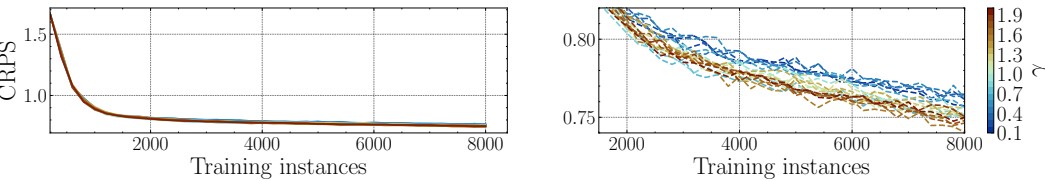

Figure 9: Continuous ranked probability score with increasing training instances for different model runs with the corresponding uncertainty measure specified by $\gamma$, averaged across three runs. The left panel shows the full run, the right panel shows a close-up.

So far, we have utilized the median heuristic as a choice of $\gamma$ for the kernel score, which in this case leads to $\gamma \approx 0.8$. However, for spatial tasks, this hyperparameter can have a substantial impact on the uncertainty estimate, as the bandwidth $\gamma$ essentially controls the sensitivity to small/large distances. Figure 4 already highlights that the kernel score shows good performance on the active learning task for predicting surface temperature for different stations. We now want to analyze whether the performance of $S_{k_\gamma}$ can be further improved by tuning $\gamma$.

Figure 9 shows that while all values of $\gamma$ seem to perform quite similarly, small improvements can be obtained by increasing the bandwidth to around $\gamma = 2$. This indicates that the optimal kernel score considers larger spatial dependencies (as compared to the one chosen by the median heuristic). This fits the underlying task, as the temperatures can be highly correlated even for larger spatial distances, for example, when both stations are affected by the same weather regime (e.g. same coastline).

