# OpenReview forum: "Uncertainty Quantification for Regression: A Unified Framework based on Kernel Scores"
_ICLR.cc/2026/Conference — Submitted to ICLR 2026_

### Official Review · Reviewer_RG8j · 2025-10-19

**Soundness:** 2
**Presentation:** 3
**Contribution:** 2
**Rating:** 2
**Confidence:** 4

**Summary:**

This paper proposes a framework for quantifying uncertainty in regression models using kernel scoring rules. It aims to provide a principled way to separate total, aleatoric, and epistemic uncertainty by generalising existing measures such as variance and entropy through kernel-based formulations. The authors introduce two estimators: one based on Bayesian model averaging and another on pairwise comparisons, and show how different kernel choices affect properties like robustness and sensitivity.

**Strengths:**

The strengths of this paper are:
- The paper elegantly connects different uncertainty measures (variance-, entropy-, and energy-based) under the general framework of kernel scoring rules, offering a consistent mathematical view of total, aleatoric, and epistemic uncertainty.
- The decomposition into total, aleatoric, and epistemic components is clearly formalised.
- I like the justification that uncertainty metrics should align with proper scoring rules!

**Weaknesses:**

The weaknesses of this paper are:
- I am left confused, in definition 4.1 the authors state that the framework assumes a non-negative kernel. But then in section 5, the Gaussian kernel is negative. Why does this clash exist in the paper? Have I missed something?
- A key weakness of the paper is the narrow scope of its empirical evaluation. All comparisons are made between different scoring rules (S_SE, S_ES, S_log, and Energy Score) applied to the same underlying ensemble models, meaning the study tests only uncertainty measures rather than fundamentally different uncertainty estimation methods. The absence of broader baselines is poor for this venue.
- Similarly, only evaluating upon two benchmarks is poor for this venue.

**Questions:**

- In Definition 4.1, the framework assumes a non-negative kernel, yet Section 5 defines the Gaussian kernel as negative. Could the authors clarify this apparent contradiction? Is the Gaussian example consistent with the theoretical assumptions on conditional negative definiteness and properness of the score?
- The empirical section compares only different scoring rules applied to the same ensemble predictor. Why were no other uncertainty estimation methods?
- How well does the proposed framework generalise to other domains such as high-dimensional vision, non-stationary time series, or real-world safety-critical regression problems?

---

> ### Author Response · Authors · 2025-11-24
>
> [1/2]
>
> We thank Reviewer RG8J for the detailed and constructive feedback. Below, we address the raised points. All clarifications and additions have been incorporated into the revised manuscript (see also our summary of revision changes).
>
> ## Weaknesses
>
>  1. **I am left confused, in definition 4.1 the authors state that the framework assumes a non-negative kernel. But then in section 5, the Gaussian kernel is negative. Why does this clash exist in the paper? Have I missed something?**
>
> Thank you for pointing out this subtle but important distinction. We have now clarified this more explicitly in the manuscript:
> Following the convention in [1], kernel scores are formally defined for *negative definite* kernels. This includes, for example, the Riesz kernel $k(x,x') = \|x-x'\|^\beta, \beta \in (0,2)$, which gives rise to the energy score. By contrast, the Gaussian kernel $k_{\gamma}$ is *positive* definite, and thus must be negated to obtain a valid negative definite kernel. Accordingly, in Section 5, we use $k = - k_{\gamma}$ [1]. With this, the definition of the kernel score is consistent with the theoretical assumptions of negative definiteness and properness of the score. However, as you pointed out correctly, we still have a mismatch between the assumed nonnegativity and the Gaussian kernel (which is negative).
>
> In section 4, we assume *without loss of generality* that $k(x,x') \geq 0$, as this guarantees that the corresponding kernel score and its uncertainty measures are also nonnegative. However, this is not a restriction, as any kernel score $S_{k'}$ can be expressed via another (strongly equivalent) kernel score $S_{k}$ ,with nonnegative kernel $k$ (compare [1], Definition 2.1 and Remark 25). Using a nonnegative representative is convenient because it guarantees that both the score and its induced uncertainty measures remain nonnegative. It does not restrict the class of admissible kernels.
> As an example, consider the (negative) Gaussian kernel, which leads to the following kernel score:
>
> $$S_{k} = \frac{1}{2} - \frac{1}{2} \mathbb{E}\left[ -\exp\left(- \frac{\|X-X'\|^2}{\gamma^2} \right) \right] + \mathbb{E}\left[- \exp \left( - \frac{\|X-y\|^2}{\gamma^2} \right) \right],$$
>
> for which it holds that $S_{k} \geq 0$. This score can equivalently be expressed using a suitable non-negative kernel. We have added a footnote to clarify this point at the definition of kernel scores.
>
>
> 2. **A key weakness is the narrow scope of its empirical evaluation [...]. The absence of broader baselines is poor for this venue. [...] Similarly, only evaluating upon two benchmarks is poor for this venue.**
>
> We agree that the empirical evaluation benefits from broader coverage. In the revised submission, we substantially expanded both the benchmark suite and the set of baselines.
>
> Therefore, we include several more benchmarks for the active learning and OOD experiments, including a high-dimensional vision task of depth regression. Similarly, apart from the already used deep ensemble and deep evidential regression we add three more baselines, including some adjusted for the high-dimensional setting. Specifically, we use the CRPS-ENS method of [3], a mixture density network [3], as well as deep evidential regression [4] and a multivariate Gaussian regressor (similar to [5]), combined with MCDropout [6] to generate the second-order distribution.
>
> Further details are provided in our revision summary. We believe these additions considerably strengthen the empirical evaluation and better align the paper with the venue’s expectations.

---

> ### Author Response · Authors · 2025-11-24
>
> [2/2]
>
> ## Questions:
>
> 1. **In Definition 4.1, the framework assumes a non-negative kernel, [...].**
>
> Please see the answer to the corresponding weakness above.
>
>
> 2. **The empirical section compares only different scoring rules applied to the same ensemble predictor. Why were no other uncertainty estimation methods?**
>
> For the T2M task in 6.1 we use both, an ensemble predictor, as well as the deep evidential regression model [4], which uses a second-order distribution, not based on ensembles. For the T2M task in 6.3, we used the ensemble predictor, as brought forward by the authors [2].
> To broaden the range of uncertainty estimation methods, we now also include a mixture density network [3], the CRPS-ENS method of [3] for the (new) UCI active learning task. For the (new) depth estimation task, we also use the (multivariate) CRPS-ENS method of [3], deep evidential regression [4], and a multivariate Gaussian regressor (similar to [5]), combined with MCDropout [6] to generate the second-order distribution.
>
> These additions cover a diverse set of predictive families and training principles, enabling a more comprehensive assessment of the proposed uncertainty measures across different neural network architectures and modeling assumptions.
>
>   3. **How well does the proposed framework generalise to other domains such as high-dimensional vision, non-stationary time series, or real-world safety-critical regression problems?**
>
> The framework is, in principle, fully general: all uncertainty measures are defined for multivariate predictive distributions and can also be applied via aggregations of marginal scores (Appendix B). Furthermore, as mentioned in section 4, by using specific kernels, one can also define the uncertainty measures on domains such as graphs or function spaces.
> To demonstrate practical applicability beyond low-dimensional settings, we evaluate our method on a real-world regression task—surface temperature prediction—which is safety-critical in contexts such as forecasting and mitigating heatwaves, such as those recently observed in Europe [7, 8]. The framework performs well for both OOD detection and active learning on this task.
> To further assess generalization to high-dimensional domains, we have added a depth-regression vision benchmark in the revised submission. This experiment confirms that our approach transfers directly to high-dimensional imaging data and yields competitive OOD-detection performance. Additional details can be found in our revision summary.
>
>
> **References:**
>
> [1] Waghmare, Kartik, and Johanna Ziegel. “Proper Scoring Rules for Estimation and Forecast Evaluation.” arXiv:2504.01781. Preprint, arXiv, April 2, 2025.
>
> [2] Feik, Moritz & Lerch, Sebastian & Stühmer, Jan. (2024). Graph Neural Networks and Spatial Information Learning for Post-Processing Ensemble Weather Forecasts. 10.48550/arXiv.2407.11050.
>
> [3] Kelen, Domokos M., Ádám Jung, Péter Kersch, and Andras A. Benczur. “Distribution-Free Data Uncertainty for Neural Network Regression.” Paper presented at The Thirteenth International Conference on Learning Representations. October 4, 2024.
>
> [4] Amini, Alexander & Schwarting, Wilko & Amini, Ava & Rus, Daniela. (2019). Deep Evidential Regression. 10.48550/arXiv.1910.02600.
>
> [5] Muschinski, T., Mayr, G.J., Simon, T., Umlauf, N., Zeileis, A., 2024. Cholesky-based multivariate Gaussian regression. Econometrics and Statistics 29, 261–281.
>
> [6] Gal, Y., Ghahramani, Z., n.d. Dropout as a Bayesian Approximation:  Representing Model Uncertainty in Deep Learning.
>
> [7] Lhotka, O., & Kyselý, J. (2022). The 2021 European heat wave in the context of past major heat waves. Earth and Space Science, 9, e2022EA002567.
>
> [8] Janoš, T., Quijal-Zamorano, M., Shartova, N. et al. Heat-related mortality in Europe during 2024 and health emergency forecasting to reduce preventable deaths. Nat Med (2025).

---

### Official Review · Reviewer_NoVr · 2025-10-29

**Soundness:** 3
**Presentation:** 3
**Contribution:** 3
**Rating:** 6
**Confidence:** 4

**Summary:**

The paper presents a theoretical framework for uncertainty estimation in regression based on proper scoring rules induced by kernels. The authors show a connection between the convex order of the respective distributions, kernel properties, and certain properties of the underlying uncertainty measures. Overall, the work provides clear explanations and a rigorous framework for constructing principled uncertainty estimates. It offers an original perspective on uncertainty, as prior works have explored related ideas of constructing scores from proper scoring rules but have not addressed the regression setting. Such a framework is indeed needed and could have significant practical applications. The authors conduct extensive experiments to validate their proposed approach and explore different application scenarios, covering several interesting use cases.

**Strengths:**

1. The work presents a solid theoretical framework and addresses an interesting problem that could have intriguing implications for downstream applications. It also removes previous limitations related to uncertainty scoring in classification tasks.
2. The mathematical soundness of the paper is strong. The propositions, results, and connections to prior work are solid.
3. The authors did a good job conducting extensive experiments to evaluate the proposed functions and covering various use cases in their experimental
4. Overall, the text is very clear and well written.

**Weaknesses:**

Overall, the paper represents solid work; however, I have a few minor remarks.
1. There is a missing space on line 147 in “However,since.”
2. The begging of Part 5 is interestingly formulated. Personally, the notation on lines 232–239 introduced some confusion for me. Why not first define the convex order between any two measures and then incorporate it directly into the proposition? For example, explicitly state in Proposition 5.1 that $Q_1 \leq_{cvx}^2 Q_2$ and that $P_1 \leq_{cvx} P_2$. Or maybe make it explicit that $P_1, P_2, Q_1, Q_2$ from introduction of the notations are re-used in the propositions later.
3. I find it somewhat peculiar that you first claim the heuristic for selecting the bandwidth is sufficient and yields good empirical results for the Gaussian kernel, yet later perform an experiment to tune the bandwidth to better align with the active learning objective.

**Questions:**

1. You mention in your work that a similar approach based on Bregman divergences has been used to construct scoring rules for classification tasks. It appears that using kernels restricts the class of measures that can be produced (you note that the log-score is not a kernel score). What, then, is the advantage of using kernels compared to Bregman divergences? Could the approach based on Bregman divergences be adapted to the same use case?

---

> ### Author Response · Authors · 2025-11-24
>
> We thank Reviewer NoVr for the detailed and constructive feedback. Below, we address the raised points. All clarifications and additions have been incorporated into the revised manuscript (see also our summary of revision changes).
>
> ## Weaknesses
>
> 1. **There is a missing space on line 147[...].**
>
> Thanks for noticing. We fixed the corresponding part.
>
> 2. **[...] Why not first define the convex order between any two measures and then incorporate it directly into the proposition? [...]**
>
> Thank you for the suggestion. We agree that introducing the convex order upfront improves readability. In the revision, we first define the convex order in full generality and then state the corresponding assumption explicitly whenever required in the propositions.
>
>
> 3. **I find it somewhat peculiar that you first claim the heuristic for selecting the bandwidth is sufficient and yields good empirical results for the Gaussian kernel, yet later perform an experiment to tune the bandwidth to better align with the active learning objective.**
>
> We appreciate the reviewer’s observation. While the median heuristic performed well across our experiments, we investigated whether the Gaussian kernel bandwidth could be further improved, particularly because $\gamma$ controls sensitivity to spatial distances. Since our prediction task is defined on a spatial domain, exploring how $\gamma$ influences the active-learning objective provides additional insight into kernel selection for different data scenarios.
>
> To clarify this point, we now (i) report the median-heuristic value for the task, (ii) discuss its interpretation in spatial settings, and (iii) compare against other uncertainty measures (log, variance, CRPS). Following reviewers feedback, we have moved the bandwidth-tuning experiment to the appendix as an ablation study.
>
> ## Questions
>
>
>
> 1. **[...] What, then, is the advantage of using kernels compared to Bregman divergences? Could the approach based on Bregman divergences be adapted to the same use case?**
>
> Proper scoring rules and Bregman divergences are closely related, but the terminology is not always consistent. Formally, proper scoring rules generalize Bregman loss functions to probability distributions [1]. Every scoring rule induces an entropy and divergence, but these need **not** be Bregman. A Bregman score [2] is specifically a scoring rule whose induced divergence is a Bregman divergence; this class includes, for example, the log-score and Tsallis scores, but **not** kernel scores.
>
> Both Bregman scores and kernel scores impose structural restrictions, and Bregman scores could in principle also be used to define uncertainty measures. However, kernel scores offer advantages central to our framework: they are closely linked to MMDs and probability metrics, admit unbiased estimators, and yield uncertainty measures with useful analytical properties. Our results rely on these kernel-specific correspondences, which would not hold for arbitrary scoring rules and would need to be analyzed specifically for the chosen Bregman score.
>
>
>
> **References:**
>
> [1] Ovcharov, E.Y., 2018. Proper scoring rules and Bregman divergence. Bernoulli 24.
>
> [2] Dawid, A.P., Musio, M., 2014. Theory and Applications of Proper Scoring Rules. METRON 72, 169–183.

---

### Official Review · Reviewer_9q6x · 2025-10-31

**Soundness:** 3
**Presentation:** 3
**Contribution:** 2
**Rating:** 2
**Confidence:** 4

**Summary:**

The authors introduce a unified framework for quantifying total, aleatoric, and epistemic uncertainty in regression tasks. The framework is built upon proper scoring rules, with a specific focus on kernel scores. By selecting different kernels, the framework can instantiate various uncertainty measures (e.g., based on squared error, energy score, or Gaussian kernels) with controllable properties such as robustness and translation invariance. The paper provides a theoretical analysis of these measures and their properties, and validates them on several downstream tasks, including out-of-distribution detection and active learning.

**Strengths:**

*   The paper tackles the important and less-explored problem of principled uncertainty quantification for regression
*   The proposed framework, based on kernel scores, is elegant
*   The theoretical connection between kernel properties (e.g., boundedness, translation invariance) and the behavior of the resulting uncertainty measures (e.g., robustness via influence functions, ordering properties) is a valuable contribution
*   The experiments are comprehensive and well-designed, covering qualitative out-of-distribution (OOD) assessment, a quantitative robustness analysis, and an interesting study on  active learning

**Weaknesses:**

My main concerns are regarding the positioning of the work with respect to recent literature and some aspects of the experimental evaluation.

*   **Related Work:** The authors seem to have missed the highly related work of Gruber & Buettner (ICML 2024), who introduce a bias-variance-covariance decomposition of kernel scores to assess generative models. While the application domain is different (generative models vs. regression), the core idea of using kernel scores to derive uncertainty measures is very similar. Gruber & Buettner derive a "kernel entropy" and a "distributional variance" from their decomposition, which are conceptually analogous to the aleatoric (AU) and epistemic (EU) uncertainty measures proposed here. Specifically, this paper's AU is the expectation of the kernel entropy H_k over the posterior, and its EU is the expected pairwise MMD, which is a measure of spread related to Gruber & Buettner's distributional variance. A thorough discussion and comparison to this work is crucially missing to properly situate the paper's contribution. How do the authors see their framework (based on the entropy/divergence decomposition of scoring rules) relating to the bias-variance decomposition of the kernel score itself?

*   **Experimental Evaluation:**
    *   Regarding the robustness analysis in Section 6.2, the experimental setup seems somewhat indirect. An outlier is introduced by corrupting the training target of one ensemble member . This conflates the training dynamics with the intrinsic robustness of the uncertainty measure at test time. A more direct evaluation would be to introduce an outlier prediction into a trained ensemble at test time (e.g., a Gaussian with a very large variance). Furthermore, the analysis is limited to aleatoric uncertainty (AU). It would be important to also analyze the robustness of epistemic uncertainty (EU), which is arguably more sensitive to model disagreement.
    *   In the task adaptation experiment (Section 6.3, Figure 4), the paper shows that larger gamma values for the Gaussian kernel score lead to better active learning performance . This is an interesting result, but it lacks interpretation. A larger gamma makes the kernel less sensitive to small distances. Why is this beneficial for this specific active learning task? Is there an optimal gamma, or does performance saturate? A more in-depth discussion on the link between the kernel hyperparameter and the downstream task performance would strengthen this section.
    *   The qualitative assessment in Figure 2 is illustrative, but the claims about the superiority of kernel-based measures for OOD detection are subjective. The color scales for the different measures are not standardized, making a fair visual comparison difficult. Could the authors provide a quantitative comparison, for instance, by computing the correlation between EU and the land-sea mask?

*   **Minor Points:**
    *   The paper states that the pairwise estimator (P) is used for the experiments because it "admits closed-form solutions" . However, it also comes at a higher O(M^2) cost. A brief justification for why this trade-off is acceptable or necessary for the conducted experiments would be helpful.
    *   The choice of the Gaussian kernel bandwidth gamma is presented inconsistently. In the introduction to Section 6, the median heuristic is mentioned as working well, but in Section 6.3, gamma is presented as a key tunable parameter. Could the authors clarify the recommended practice? Was the median heuristic used for the experiments in 6.1 and 6.2?

**Questions:**

See above

---

> ### Author Response · Authors · 2025-11-24
>
> [1/3]
>
> We thank Reviewer 9q6x for the detailed and constructive feedback. Below, we address the raised points. All clarifications and additions have been incorporated into the revised manuscript (see also our summary of revision changes).
>
> ## Related work:
>
> **The authors seem to have missed the highly related work of Gruber & Buettner (ICML 2024), who introduce a bias-variance-covariance decomposition of kernel scores to assess generative models. [...] How do the authors see their framework (based on the entropy/divergence decomposition of scoring rules) relating to the bias-variance decomposition of the kernel score itself?**
>
> We thank the reviewer for pointing out the highly relevant work of Gruber & Buettner (2024) [4]. We indeed missed it and now include it in the related work (lines 497-500) and contributions (line 64), together with an explicit discussion of similarities and differences.
>
> Gruber & Buettner (2024) develop a bias–variance–covariance decomposition of kernel scores for generative models. Their “kernel entropy’’ and “distributional variance’’ are conceptually aligned with our AU and EU: kernel entropy aligns with our aleatoric term, and distributional variance with our expected pairwise divergence (MMD-type) term. Our framework, however, does not introduce a new decomposition; it builds on the established entropy/divergence decomposition of proper scoring rules [3,5] (also referenced by Gruber & Buettner (2024)) and extends it to multivariate regression. The main theoretical novelty lies in proving how specific kernel properties (such as translation invariance and robustness) translate into properties of AU/EU in regression, and in providing a unified view in which variance-, entropy-, and kernel-based measures arise as special cases.
>
> The empirical evaluation settings also differ: Gruber & Buettner (2024) focus on generative modeling with purely sample-based estimators and instance-level correlations with loss, while we study probabilistic regression (including parametric models and diffusion ensembles), where closed-form expressions can be available and AU/EU are explicitly separated and evaluated on tasks such as OOD detection, robustness, and active learning. Thus, while both works highlight the usefulness of kernel scores for uncertainty quantification, we believe the contributions are complementary: Gruber & Buettner provide a novel decomposition for generative models, whereas we extend and unify scoring-rule decompositions from classification to regression and analyze how kernel properties shape AU/EU behavior in this setting, both theoretically and empirically.

---

> ### Author Response · Authors · 2025-11-24
>
> [2/3]
>
> ## Experimental Evaluation
>
> 1. **Regarding the robustness analysis in Section 6.2, the experimental setup seems somewhat indirect. [...] Furthermore, the analysis is limited to aleatoric uncertainty (AU). It would be important to also analyze the robustness of epistemic uncertainty (EU), which is arguably more sensitive to model disagreement.**
>
> We appreciate the reviewer’s observation and agree that robustness deserves a clearer justification. A direct manipulation of an ensemble member at test time—such as injecting a prediction with extremely large variance—is conceptually aligned with our theoretical robustness analysis in Appendix C.2, which explicitly studies the effect of perturbing a single ensemble component. This theoretical result already characterizes the expected behavior under such test-time outliers.
>
> For this reason, our experiments focus on a complementary scenario: introducing an outlier during training to evaluate how each uncertainty measure responds when one ensemble member is systematically corrupted. Although the mechanism differs, the resulting behavior is consistent with the theoretical predictions.
>
> Initially, we restricted the evaluation to aleatoric uncertainty (AU). We agree that epistemic uncertainty (EU), being more sensitive to model disagreement, provides an equally important perspective. Since an outlier ensemble member should, by definition, increase EU even for an ideal measure, interpreting EU is more difficult. Nevertheless, following the reviewer’s suggestion, we have added a the results for EU to the main text (Table 2) and to the appendix (Table 5 and Figure 7). Similar to AU, the bounded kernel score is least affected, due to it being bounded. However, for EU the the log-score and squared-error measures diverge similarly, while the CRPS-based measure grows more slowly with increasing distortion.
>
> 2. **In the task adaptation experiment (Section 6.3, Figure 4), the paper shows that larger gamma values for the Gaussian kernel score lead to better active learning performance. This is an interesting result, but it lacks interpretation. [...]**
>
> Thank you for highlighting this important point. We expanded the analysis of the Gaussian-kernel bandwidth in Appendix C.4. Empirically, the optimal $\gamma$ is slightly larger than the median-heuristic value, which is consistent with the structure of the T2M task: temperatures remain strongly correlated even across larger spatial distances—for example when stations share the same weather regime (e.g., coastal influences). A larger $\gamma$ therefore captures these longer-range dependencies more effectively.
>
> Following the reviewers’ suggestions, we also added comparisons with the other uncertainty measures in the main experiment and moved the bandwidth study to Appendix C.4, where it is treated as an ablation showing how performance can be improved beyond the default heuristic.
>
>
> 3. **The qualitative assessment in Figure 2 is illustrative, but the claims about the superiority of kernel-based measures for OOD detection are subjective. [...] Could the authors provide a quantitative comparison, for instance, by computing the correlation between EU and the land-sea mask?**
>
> We agree that the qualitative visualization alone is insufficient and that differing color scales complicate a fair comparison. In the revision, we therefore add a quantitative evaluation: specifically, we report AUROC values distinguishing ID vs. OOD regions, allowing a standardized comparison across all uncertainty measures. In addition, we include a new high-dimensional vision task and report the same metric there. Together, these quantitative results provide a clearer and more robust assessment of the relative performance of the different uncertainty measures.

---

> ### Author Response · Authors · 2025-11-24
>
> [3/3]
>
> ## Minor points
>
> 1. **The paper states that the pairwise estimator (P) is used for the experiments because it "admits closed-form solutions" . However, it also comes at a higher O(M^2) cost. A brief justification for why this trade-off is acceptable or necessary for the conducted experiments would be helpful.**
>
>
> We agree that the computational trade-off warrants clarification. In our setting, using the pairwise estimator (P) is justified for two reasons. First, for the BMA estimator (B), closed-form expressions generally do not exist for several of the considered measures—including the log-score, CRPS, and Gaussian-kernel score. Relying on (B) would therefore require additional numerical estimators, which may be costly or unavailable (e.g., for the log-score). In contrast, when closed-form expressions for (P) are available—as in the case of evidential regression—no sampling is required, avoiding the $\mathcal{O}(M^2)$ cost.
>
> Second, practical ensemble-based uncertainty estimation typically uses relatively small ensemble sizes (M \approx 10), making the quadratic overhead manageable in all experiments considered. Finally, since the uncertainty estimators are oftentimes not incorporated into training (for example, in OOD detection), the computational complexity usually plays a negligible role as compared to the training cost. For these reasons, the pairwise estimator provides the most reliable and tractable choice for our experiments.
>
> 2. **The choice of the Gaussian kernel bandwidth gamma is presented inconsistently. In the introduction to Section 6, the median heuristic is mentioned as working well, but in Section 6.3, gamma is presented as a key tunable parameter. Could the authors clarify the recommended practice? Was the median heuristic used for the experiments in 6.1 and 6.2?**
>
>
> That is a good point, which has also been raised by other reviewers and which we understand can raise some confusion. Indeed, we used the median heuristic for $\gamma$ in all experiments in Sections 6.1 and 6.2, and we have clarified this explicitly in the revised manuscript. The tuning experiment in Section 6.3 was not meant to present $\gamma$ as a key hyperparameter but rather to illustrate that performance can be improved beyond the simple heuristic, particularly in spatial prediction settings where correlations persist over larger distances.
>
> To avoid confusion, we (i) clearly state the use of the median heuristic in the main text, (ii) include the heuristic value as a baseline in the tuning experiment, (iii) discuss the interpretation of $\gamma$ for spatial problems, and (iv) move the tuning procedure to Appendix C.4 as an ablation study, following reviewer suggestions. We also added comparisons to the log-score, variance, and CRPS for the active learning task, to offer a ground for comparison.
>
>
> **References:**
>
> [1] Hofman, P., Sale, Y., Hüllermeier, E., Uncertainty Quantification with Proper Scoring Rules: Adjusting Measures to Prediction Tasks, arXiv preprint arXiv:2505.22538
>
> [2] Nikita Kotelevskii, Vladimir Kondratyev, Martin Takác, Eric Moulines, and Maxim Panov. From risk to uncertainty: Generating predictive uncertainty measures via bayesian estimation. In The Thirteenth International Conference on Learning Representations, 2025
>
> [3] Gneiting, T., Raftery, A.E., 2007. Strictly proper scoring rules, prediction, and estimation. Journal of the American Statistical Association 102, 359–378.
>
> [4] Gruber. S., Buettner, F., A Bias-Variance-Covariance Decomposition of Kernel Scores for Generative Models, 41st International Conference on Machine Learning, 2024.
>
> [5] Dawid, A.P., 2007. The geometry of proper scoring rules. AISM 59, 77–93.

---

### Official Review · Reviewer_tJPo · 2025-11-01

**Soundness:** 2
**Presentation:** 2
**Contribution:** 2
**Rating:** 2
**Confidence:** 4

**Summary:**

There is a large amount of literature on how to obtain second-order distributions that represent estimated epistemic, aleatoric, and total uncertainty, which is still an active field of research (A). The submission follows a different path (B): Instead of improving second-order distributions, it studies uncertainty measures that summarize the estimated second-order distribution into three scalar values EU(x), AU(x), and TU(x). The submission argues that this summarization step (B) obtained much more attention for classification than for regression. Therefore, this submission focuses on studying this step (B) for regression. There are multiple different uncertainty measures available for this step (B), and the submission unifies these uncertainty measures, studies them theoretically and experimentally, and suggests ways to design new approaches for this step (B).

Figure 1 nicely shows how these different uncertainty measures can be different in a synthetic example.

The abstract and the conclusion are framed as if studying the difference of these complexity measures from step (B) has a significant impact on downstream performance, and as if the paper provides useful guidelines for task-specific uncertainty measures for practitioners. However, from my perspective, all the experiments on real-world data rather convey the opposite message: For the real-world experiments in this submission, there is no substantial difference in downstream performance across the different complexity measures in step (B). Therefore, the choice of uncertainty measure seems rather irrelevant for downstream performance in practice.

This mismatch between claims and experiments leads me to a score of 2-3.

If you more honestly reformulated the paper, that no major effect of the choice among these uncertainty measures on downstream performance was found, I would raise it to 3-4.

If you significantly extend the number of datasets to show reliably among multiple different datasets that the choice among these uncertainty measures can reliably be ignored, I would raise it to 3-5, depending on the quality/rigor/quantity of the experiments. This could encourage the community to keep focusing on (A) rather than on (B).

If you show that the negligible effect of the choice among these uncertainty measures on downstream performance for the considered datasets was rather an exception, but for many other datasets/down-stream tasks the choice among these complexity measures is really important for downstream performance, then I would raise my score to 3-7 (depending on how big the difference is, how extensive and convincing the experiments are [quality, rigor, number of experiments, number of datasets, number of downstream tasks, ablation studies], and depending on how practical the guidelines are how to pick the right uncertainty measure in practice). But to be honest, I think this is outside the scope of this revision, but rather for a future resubmission. It would be a very interesting result if you manage to show that (B) is (almost) as important as (A), while the majority of the literature focuses on (A). However, the current experiments do not hint towards this direction.

(These scores are based on the assumption that the mathematical mistake will be fixed during the revision.)

**Language-wise clarification:**

The submission clearly distinguishes between two different tasks:

* (A) Obtaining a representation of uncertainty via second-order distributions, and
* (B) summarizing a second-order distribution into three scalar values EU(x), AU(x), and TU(x).

This distinction is explained well, very reasonable and very clear. However, I don’t think that the terms used to describe them are in accordance with the literature:

The submission refers to (A) as “uncertainty representation”, and (B) as “uncertainty quantification”. I think a big part of the literature refers to “uncertainty representation” as a general umbrella term that also includes (A). For example, Wikipedia says: “Uncertainty quantification (UQ) is the science of quantitative characterization and estimation of uncertainties in both computational and real-world applications. It tries to determine how likely certain outcomes are if some aspects of the system are not exactly known.” I think a second-order distribution is also a quantitative characterization of uncertainty, and second-order distributions try to determine how likely certain outcomes are if some aspects of the system are not exactly known.

For me, an uncertainty representation is a language to express or communicate uncertainty. One representation of uncertainty is a second-order distribution; another representation of uncertainty is the 3 numbers EU(x), AU(x), and TU(x), and there are other representations of uncertainty, e.g., via sets instead of distributions. To my understanding, (B) is about translating/summarizing one representation of uncertainty (concretely, second-order distributions conditioned on x) into another representation of uncertainty (concretely, 3 numbers EU(x), AU(x), and TU(x)).

So in some sense one could argue also that the naming should be done exactly the other way around: (A) to which the paper refers as “uncertainty representation” is also about “uncertainty quantification” and (B) to which the paper refers to as “uncertainty quantification” is about changing the “uncertainty representation”.

Within the submission, (A) is clearly and consistently referred to as “uncertainty representation”, and (B) as “uncertainty quantification”. However, it would be nice to use different (maybe new) terms that are not that overloaded yet. Or at least write explicitly that you are following a specific subculture of the literature that uses the terms in this way, while explicitly acknowledging that there are other subcultures in the literature that use these terms differently.

**Strengths:**

Aleatoric, Epistemic, and total uncertainty are important topics for many applications, which require further research.

I like the idea of a simple post-processing step for an already trained model to potentially improve the downstream performance. Figure 1 nicely visualizes why changing the uncertainty measure could theoretically be such a step with theoretically high potential for impact. I totally see that it was very well motivated to start this project to find out if this can actually have a significant impact on downstream tasks. If clear evidence was found during this project that this choice is actually impactful, this would have been a much stronger paper. However, I think there is also value in publishing “negative” results about theoretically promising ideas that have not (yet) turned out to be particularly useful for practice.

Figure 1 provides good intuition about the mathematical difference between these uncertainty measures.

The paper provides a nice overview of different uncertainty measures by explaining how multiple different uncertainty measures can be seen as special cases of a general formalism. (If I understand it correctly, none of these formalisms are new, however?) (I am not familiar enough with the literature to judge how complete this overview is.)

The theoretical results on certain properties of complexity measures seem plausible. However, I did not check the proofs, and I think the novelty of these results is overstated due to a mathematical mistake in the argument.

The main idea of the experimental setup of Section 6.3 is reasonable.

The potential to extend this beyond regression to other output modalities by using appropriate kernels seems interesting. However, no experiments were conducted in this direction.

**Weaknesses:**

**1 Major weakness: The mismatch between the overall storyline and the actual experimental results on real-world data.** The overall storyline is formulated as if (B) is very important for downstream performance (while the majority of the literature only focuses on (A)). However, I don’t see any big impact of (B) on any downstream task in this submission. Let’s go together through all the experiments:

1a Experiment of Section 6.1 QUALITATIVE ASSESSMENT OF UNCERTAINTY QUANTIFICATION

I think you are interpreting too much into Figure 2: When I look at the four different plots of AU for different choices of step (B), I see 4 qualitatively extremely similar plots. Each of these plots shows more aleatoric uncertainty in high-altitude regions and lower aleatoric uncertainty in flat regions. IF you compare Figure 2 to Figure 6 in the appendix, then the impact of step A (the difference between Figure 2 and 6 and the difference for different lamdas) seems much larger than the impact of step (B) in Figure 2. In Figure 6 in the appendix, step (B) seems to make a difference too, but it seems to be in favor of S_log, which is kind of the default in the literature and cannot be expressed via Kernel Scores, so it does not really motivate why one should study Kernel Scores.

Also, for the EU, all 4 uncertainty measures in Figure 2 behave qualitatively similarly in the sense that they all assign higher epistemic uncertainty to the water than to the land. The scales are differently distorted. Maybe taking the log of the log for the color scale of the EU for S_log would make the picture look more similar to the others.

Furthermore, I criticize that while a pretty plot for 1 dataset is very nice, it is not sufficient to make general claims. The aleatoric uncertainty in Figure 2 nicely aligns with your intuition that in more mountainous areas, the aleatoric uncertainty should be higher. However, without quantitatively measurable performance metrics for relevant downstream tasks, this qualitative assessment of the plot is not enough, especially when the qualitative assessment is so tight as here. I don’t really see which of these uncertainty measures is better or worse in Figure 2.

1b Experiment of Section 6.2 ROBUSTNESS ANALYSIS

This experiment seems quite artificial. In practice, there is no reason to add artificial noise to the training data of one ensemble member. For practically used ensemble members, there is usually some model selection step where poorly performing ensemble members are sorted out. This would already eliminate the distorted ensemble. I do like the experiment, as it shows a clear result that is exactly as theoretically expected. But is this a common scenario in real-world applications that one of the ensemble members is outlying in this specific way? When people talk about robustness, they often think about outliers in the training data, which can occur in practice, but how often do you have an outlying ensemble member? Real-world examples of this would be a valuable addition.

This experiment also misses any downstream performance. Do I want the outlying model ensemble to have an influence on the estimated aleatoric uncertainty, or not, to improve downstream performance, if the outlier was not artificially added but might actually carry some important information? How do I decide for which downstream tasks I want this type of robustness for and which I don’t? What kind of downstream tasks are affected by this? How are they affected by this? How much? I like this experiment, but I don’t think that it is sufficient to claim that step (B) is particularly important for downstream tasks.

1c Experiment of Section 6.3 TASK ADAPTATION OF MEASURES

This is the only section with experiments on downstream tasks. However, it only studies 1 dataset, and for this dataset, step (B) seems to be quite irrelevant for downstream performance. There are two subexperiments, one in Figure 3 and one in Figure 4.

Figure 3:

Line 401-402 correctly describes “relatively minor variation between individual measures” as the different uncertainty measures from step (B) perform visually indistinguishably. At least for these specific downstream tasks, I would say that the choices made in step (B) are basically irrelevant. Figure 3 shows a much stronger difference for the choice of loss functions during training (i.e., step (A)).
In addition to Figure 3, some quantitative performance metrics would be helpful. E.g., in the spirit of abstention (http://arxiv.org/abs/2404.10960), plot on the x-axis the percentage of rejected datapoints and on the y-axis the average metric over the non-rejected datapoints: For example, for x=20%, you plot the average score of the 80% least uncertain datapoints. Then lower curves correspond to better uncertainty measures than the higher curves. This would be less noisy than the current Figure 3.

Figure 4:

Figure 4 shows a well-defined downstream task with a clearly defined performance metric. When plotted on a typical scale (i.e., the left plot), the different uncertainty measures from step (B) are visually indistinguishable. Only on the close-up one can see that, depending on the step (B), the CRP varies between 0.74 and 0.76 after 8000 training instances. This does not seem to be a big difference. It is absolutely not clear if this difference is statistically significant. (You repeated each gamma for 3 runs. Can you get statistical significance from this?) To get this difference, you need to vary gamma over quite a wide range. Do people actually use such extreme values of gamma? To me, the results look rather inconsistent: For less than 3500 training instances, gamma>1 seems to perform best; for more than 4000 training samples, gamma>1 seems to perform best. I think the statement: “The results clearly demonstrate systematic task adaption: larger values of γ consistently yield lower CRPS (highlighted by the color gradient), indicating better model performance.” strongly overstates the result. The submission only evaluated this one single dataset, and there the differences are tiny and not consistent over the number of training instances. I would summarize the same results as “The results **don’t** demonstrate clear systematic task adaptation. There might be a tendency that larger values of gamma yield lower CRPS in the case of more than 4000 training samples for this specific dataset. However, this tendency is not visible for fewer than 4000 training samples.” What should a practitioner learn from this experiment? Larger values of gamma are always better? Or gamma=1.6 is typically a good choice. Or one should always do a very expensive hyperparameter optimization (HPO) to find out which gamma works best (however, if one does the HPO based on the first 3000 samples, the resulting gamma is not particularly good for the performance after 8000 samples)? Or the choice of gamma does not really matter for downstream performance; instead, one should rather spend more time optimizing step (A). I don’t think that any of these claims is well substantiated based on this one experiment on one dataset. In the discussion, the submission claims “Our analysis highlights how specific properties of kernel scores directly translate into distinct characteristics of the induced uncertainty measures, offering practical guidance for their selection and adjustment” [Lines 468-471]. What exactly is the practical guidance? I don’t think that there are any conclusive results on this in this submission.

I would recommend you also compare with the other uncertainty measures for the experiment in Figure 4. Maybe this comparison could reveal a larger difference.

**2 Mathematical Mistakes and Overstated Novelty**

In lines 251-252, the submission claims: “However, our notion is more general, as every mean-preserving spread implies a convex order, but not vice versa.” I think the second part of this statement is wrong. I think it does also hold vice versa. Convex order is equivalent to mean-preserving spread:  https://en.wikipedia.org/wiki/Mean-preserving_spread#Relation_to_expected_utility_theory. If you think that I am wrong, please explain and/or provide a counterexample of any 2 random variables whose distributions are in convex order but cannot be expressed as a mean-preserving spread. If you want, I can also give you a more detailed mathematical proof of why I think they are equivalent.

The submission claims that one of the main novelties of the submission’s theory over the existing theory by Wimmer et al. (2023), Sale et al. (2023a) is that the submission generalizes the results from mean preserving spread to convex order [Lines 251-252 and 257]. However, I don’t agree that the submission's results are more general, since convex order is equivalent to mean-preserving spread. Therefore, Propositions 5.1 and 5.2 are not novel but rather an equivalent reformulation of the results by Wimmer et al. (2023); Sale et al. (2023a). Please correct me if you disagree.

**3. Minor Weakness:**

The usage of the terms "uncertainty representation" and "Uncertainty quantification” to distinguish between (A) and (B) is not aligned with the literature. These terms are introduced as if this specific usage of these terms was widely accepted across the literature, but it is not.

**4. Presentation issues:**

Lines 41-42: “This choice of measure is crucial, as it directly influences both the decision-making process and the performance of downstream tasks.” This is strongly overstating how crucial this is. The experiments don’t show any evidence (not even any real-world example) where the choice of uncertainty measure crucially influences the performance of any relevant downstream task.

Line 133: S(P,P) might be more concise than the integral.

Lines 147-148: “However, since the BMA distribution generally differs from the true predictive distribution, this estimator can be misleading (Schweighofer et al., 2023).” More details would be interesting here. What do you mean by “true predictive distribution”? How does BMA deviate from this?

Line 160-161: “From now on, we refer to the two different methods with index B and P for BMA and pairwise estimation, respectively.” But then you don’t use this notation anymore. Does Section 5 hold for both B and P or only for one of them?

Line 168: “negative definite kernel”. I think negative definite kernels are less known than positive definite kernels. Maybe a quick definition would be useful here, but I leave it up to you to decide if this is necessary.

Line 205: “U-statistic (Gretton et al., 2012)” might not be widely known in the ML community.

Line 213-214: “The energy score (Gneiting & Raftery, 2007)” was not defined until here. You define it 2 pages later. Maybe consider defining it directly here?

Line 244: It would be helpful to explicitly mention that you are still assuming $Q_1\leq_{cx}^2 Q_2$ here. You can simply add “for all $Q_1\leq_{cx}^2 Q_2$” here without any additional line break.

Line 252: If you actually believe that “but not vice versa” holds, then you need to explain why. I am very sure that “but not vice versa” is wrong.

Section 6.2 is hard to read, mainly because in Line 366, it was not clear to me what “predictions” you are talking about. I think by “predictions” you actually mean “estimates of the aleatoric uncertainty uncertainty measure AU”? I also think that the notation $\hat{y}$ is too overloaded in Line 359. In Line 359, I would recommend writing $\tilde{y}$ instead of $\hat{y}$. In lines 363-367, you might want to write $\hat{a}$ instead of  $\hat{y}$.

**Questions:**

Q1: Lines 222-224: “For instance, kernel scores allow for comparing (almost any) arbitrary distributions with an unbiased estimator, which can be important, for example, for mixture-of-expert models, where each expert issues a prediction in a different format.” I think many more details would be needed here to understand your thoughts on this. About what kind of mixture-of-experts (MoE) are you talking about? The same MoEs that are used in LLMs? To my understanding, each expert in a MoE, as they are used in LLMs, outputs a tensor of the same dimensions, which I would consider the same format. What do you mean by different formats? Where does this occur in practice and how?

Q2: Figure 2: You only show the AU over the land, but not over the water? Why? If so, you need to clearly specify this in the caption or at least somewhere in the text, or as an absolute minimum, somewhere in the appendix. Or does AU happen to always have the white value over the water?

Q3: In line 377, what does “task loss” refer to? Is the task loss the loss that is used as training loss and as evaluation loss? This was quite confusing when reading the submission for the first time.

Q4: In Line 398, you mention “the model”. Is it actually just one individual model or an ensemble of models? If it’s an ensemble, I would write “the ensemble” instead to avoid any ambiguity.

Q5: How do you suggest doing the HPO for Figure 4 in practice? In practice, the main motivation for active learning is that obtaining labels is very expensive. In Figure 4, you collect different labels for each different value of gamma. However, only after collecting these 8000 labels you see which gamma performed best. But once you have already collected the 8000 labels for each gamma, there is no point anymore in predicting which gamma would have been best for collecting 8000 labels. How would you imagine a pipeline that takes advantage of the different performance of different gammas?

Q6: In the Discussion, in Lines 469-472, the submission says: “Our analysis highlights how specific properties of kernel scores directly translate into distinct characteristics of the induced uncertainty measures, offering practical guidance for their selection and adjustment.”. How practical is this guidance? Can you give a practical example where this guidance practically improves a practically relevant metric of a practically relevant downstream task?

Q7: In Line 1025-1027 of Appendix C: What is the actual input to the DRNs? Is it only the mean prediction of the ECMWF integrated (ensemble) forecast system (IFS)? Or is it actually everything that is mentioned in Section C.1? In other words, is the input only the predicted temperature or all the covariates listed in Section C.1? Does ECMWF predict all the covariates listed below or only the temperature?

Q8: In Appendix C.1, Deep evidential regression seems to perform very poorly compared to the Deep ensemble. Do you agree?

Q9: Do you see any interesting connections to optimal transport (e.g., the Wasserstein metric between probability measures)? Can you also fit them into this framework?

---

> ### Author Response · Authors · 2025-11-24
>
> [1/5]
>
> We thank Reviewer tJPo for the extremely detailed and very constructive feedback. Below, we address the raised points. All clarifications and additions have been incorporated into the revised manuscript (see also our summary of revision changes).
>
> ## Major points
>
> 1. **Language-wise clarification: [...] Within the submission, (A) is clearly and consistently referred to as “uncertainty representation”, and (B) as “uncertainty quantification”. [...]**
>
> *Analyzing (A) vs (B)*:
>
> We thank the reviewer for the detailed and constructive assessment. We agree that (A) and (B) are conceptually distinct and that the paper’s framing should better reflect the relative impact of each. We have toned down corresponding sections, mainly in the introduction.
>
> We fully agree that advances in (A)—the construction of high-quality second-order distributions—often have the largest effect on downstream performance, and our experiments confirm this intuition. The goal of our paper is therefore not to argue that (B) is equally influential, but to systematically study the quantification step (B). Even when an excellent uncertainty representation is given (e.g., from an existing or pre-trained model), practitioners still require principled ways to summarize it—for OOD detection, active learning, robustness analysis, or simply interpreting the predictive uncertainty itself. Our work investigates this summarization step in a unified theoretical framework.
>
> We agree that the original abstract and conclusion overstated the downstream importance of choosing one uncertainty measure over another. We have revised both to state the empirical finding more honestly: across the datasets we study, differences between uncertainty measures in (B) are often small. At the same time, our new experiments indicate settings where kernel-based measures seem beneficial.
>
> Even if improvements in (A) typically have the strongest impact on downstream performance, there remain many practical scenarios where (B) is unavoidable and meaningful. In applied domains such as medical imaging, climate science, or engineering simulations, practitioners often cannot modify the underlying predictive model and instead must work with a fixed second-order distribution provided by a pre-trained system. In such cases, principled uncertainty summaries are essential—for OOD detection, decision-making, or interpretation of uncertainty. Our experiments show that, within this setting, some summarization methods (particularly kernel-based ones) can outperform classical choices and exhibit favorable properties such as robustness or unbiased sample-based estimation. Thus, even if (A) drives overall predictive quality, studying (B) remains important in scenarios where only the summarization step is accessible or practically relevant.
>
>
> *Uncertainty quantification & Representation*
>
> We agree that these terms are used differently across subcommunities. In this paper, we follow a convention that we have found useful and that is also adopted in parts of the literature (e.g. [7,8,9]: by uncertainty representation we mean the mathematical object used to represent (viz. encode) predictive uncertainty (e.g., a probability distribution, a credal set, a second-order distribution), whereas uncertainty quantification refers to a real-valued functional that maps such a representation to a single number (e.g., an entropy-type measure). Under this convention, two models can thus have the same uncertainty quantification while relying on different representations. We have clarified this terminology in the revised version and acknowledge that other strands of the literature may use these terms differently.

---

> ### Author Response · Authors · 2025-11-24
>
> [2/5]
>
> 2. **Major weakness: The mismatch between the overall storyline and the actual experimental results on real-world data.**
>
>
> For a general statement regarding uncertainty representation vs uncertainty quantification, please also see our answer above. Here are more specific answers to the individual experiments.
>
> *1a:*
>
> We appreciate the reviewer’s careful analysis of Figure 2 and agree that the qualitative differences between uncertainty measures are more subtle than originally emphasized. The figure should be read as an illustrative example rather than evidence for the performance of any particular kernel score. We have revised the text accordingly to avoid overinterpretation. To address the reviewer’s concern, we now add quantitative evaluations for OOD detection. Specifically, we report AUROC scores between ID and OOD regions for all uncertainty measures. In the univariate setting of Figure 2, this analysis confirms that the log-score performs best, consistent with its strong performance in prior work.
>
> We further implemented a depth regression task that allows multivariate, sample-based uncertainty evaluation (for ensemble and mixture-based methods). In this higher-dimensional setting, the kernel-based uncertainty measures show improved OOD performance, suggesting that they become particularly useful beyond simple univariate regression.
>
> Across all datasets and model parameterizations, the energy score achieves the best average rank, providing a more robust empirical picture than the single qualitative plot. Overall, the qualitative visualizations remain helpful for intuition (for example, AU increasing in mountainous regions), but the strengthened quantitative evaluation now supports a more general and reliable comparison across uncertainty measures.
>
> *1b:*
>
> We agree that the setup in Section 6.2 is artificial and not representative of most real-world ensemble workflows. The goal of the experiment, however, was not to mimic a realistic training pipeline but to provide a controlled validation of the robustness properties from Proposition 5.3. The observed growth rates match the theoretical influence-function analysis and therefore serve as a sanity check for the kernel-based measures. This is also the way section 6.2 was framed.
>
> While this experiment alone does not justify claims about the downstream importance of (B), it does demonstrate that, when (B) is of importance, kernel-based measures (with bounded kernels) offer strictly better robustness than entropy-based or variance-based alternatives.
>
> *1c:*
>
> We have revised the experiment following the reviewers’ suggestions. First, we now evaluate the post-processing task for all uncertainty measures (using the median heuristic for the kernel score). While the kernel score performs best, its performance is comparable to the energy score and log-score; only the squared-error measure performs notably worse, consistent with its violation of Proposition 5.1. We moved the detailed tuning of $\gamma$ to the appendix and clarified in the main text that $\gamma$ should be interpreted as a task-dependent adjustment of spatial sensitivity rather than a claim of consistent performance gains.
>
> To further strengthen the empirical basis, we added additional UCI datasets to the active-learning experiment. Across these datasets and predictive models, the kernel-based measure performs reliably well and achieves the best average rank overall. We refer to our main revision comment for a full summary of these updated results.
>
>
> 2. **Mathematical Mistakes and Overstated Novelty**
>
> Thank you for this correction. You are right that convex order and mean-preserving spread are equivalent. In the original text, we phrased this inaccurately by implicitly comparing convex order to a mean-preserving spread under an i.i.d.-noise interpretation rather than the correct conditional-noise formulation. We have removed the misleading sentence and corrected the surrounding discussion.
>
> Importantly, our results do not rely on claiming a generalization from mean-preserving spread to convex order. That remark was not intended as a contribution and is now omitted. The theoretical contributions of the paper lie instead in extending these ideas to regression, specifically by (i) providing a unified scoring-rule framework that treats variance-based, scoring-rule–based, and kernel-based uncertainty measures in a single formulation, (ii) showing how these measures can be derived and analyzed using proper scoring rules (including variance and kernel scores, which have not previously been connected in this way), and (iii) proving explicit correspondences between properties of kernel scores (e.g., translation invariance, robustness) and the resulting behavior of aleatoric and epistemic uncertainty in regression.

---

> ### Author Response · Authors · 2025-11-24
>
> [3/5]
>
> ## Presentation issues:
>
> **Lines 41-42: [...] This is strongly overstating how crucial this is.**
>
> Thank you for raising this point. We agree that the original phrasing may have appeared too strong in the context of our own experiments. The intention of Lines 41–42 was to summarize existing literature showing that the choice of uncertainty measure can influence downstream performance—primarily in classification settings, as demonstrated in works such as [7, 8, 9]—rather than to claim that our experiments alone establish this. To avoid overstating the implication, we have revised the wording to make clear that this statement refers to findings from concurrent work, not to evidence provided by our own evaluations.
>
> **Line 133: S(P,P) might be more concise than the integral.**
>
> Thanks for pointing that out, we agreed and changed it accordingly.
>
> **Lines 147-148: [...] What do you mean by “true predictive distribution”? How does BMA deviate from this.**
>
> Thanks for the question, we added a small explanation in the corresponding lines and give more details here.
> In principle, the true (predictive) model has parameters ${w}^{*}$ , which are however, unknown. Therefore, the BMA, which averages over all possible parameters, weighted with the posterior distribution, is used as an approximation.
>
> These two do not need to coincide everywhere, even if the BMA concentrates around the true parameter ${w}^{*}$.
>
> Furthermore, in a misspecified model, the true ${w}^{*}$ might receive zero posterior probability and every BMA predictive is necessarily different from the true predictive.
>
> **Line 160-161: “From now on, we refer to the two different methods with index B and P for BMA and pairwise estimation, respectively.” But then you don’t use this notation anymore. Does Section 5 hold for both B and P or only for one of them?**
>
> As indicated in the footnote on page 5, the results hold for both estimators. Arguably, the notation/index of B and P might not be necessary, but is used later on in the proofs, which is why we think it might be helpful to mention it.
>
> **Line 168: [...] I think negative definite kernels are less known than positive definite kernels. Maybe a quick definition would be useful here, but I leave it up to you to decide if this is necessary.**
>
> We see your point and added a brief definition and a reference for a negative definite kernel as a footnote.
>
> **Line 205: “U-statistic (Gretton et al., 2012)” might not be widely known in the ML community.**
>
> We see your point and have removed the term U-statistic entirely, to avoid confusion. Instead it now reads “[...] admit an unbiased empirical estimator (Gretton et al., 2012)”, which has the same information content as before.
>
> **Line 213-214: “The energy score (Gneiting & Raftery, 2007)” was not defined until here. You define it 2 pages later. Maybe consider defining it directly here?**
>
> Thank you for noting this. Since the definition of the energy score is not required for the discussion of homogeneity in this section, we removed the sentence at Lines 213–214. The energy score is now introduced together with its definition later in Section 5, where it is directly relevant.
>
> **Line 244: It would be helpful to explicitly mention that you are still assuming $Q\_{1} \leq Q\_{2}$. You can simply add for all [..].**
>
> We completely agree and changed the corresponding assumption (also in Proposition 5.2, which had the same problem).
>
> **Line 252: If you actually believe that “but not vice versa” holds, then you need to explain why. I am very sure that “but not vice versa” is wrong.**
>
> See also our answer on the major points and the mathematical mistakes.
>
>
> **Section 6.2 is hard to read, mainly because in Line 366, it was not clear to me what “predictions” you are talking about. I think by “predictions” you actually mean “estimates of the aleatoric uncertainty uncertainty measure AU”? I also think that the notation $\hat{y}$ is too overloaded in Line 359. In Line 359, I would recommend writing $\tilde{y}$ instead of $\hat{y}$. In lines 363-367, you might want to write $\hat{a}$ instead of $\hat{y}$.**
>
> Thank you for pointing this out. We agree that the wording and notation in Section 6.2 were unclear and that the MAPE was not defined on the appropriate quantities. Following your suggestions, we now explicitly formulate the MAPE in terms of the uncertainty measures $M_i \in \{ \mathrm{AU, EU}\}$, making clear that the robustness analysis concerns the uncertainty estimates rather than the predictions themselves. We also updated the notation accordingly, replacing the overloaded $\hat{y}$ with $\tilde{y}$.

---

> ### Author Response · Authors · 2025-11-24
>
> [4/5]
>
> ## Questions:
>
> **Q1: Lines 222-224: [...] About what kind of mixture-of-experts (MoE) are you talking about? The same MoEs that are used in LLMs? [...]**
>
> Thank you for pointing out the ambiguity. Our use of the term mixture-of-experts was not intended to refer to the MoE architectures commonly used in LLMs. We have revised the wording in the main text to avoid this confusion.
>
> Here, we refer more generally to mixtures of probabilistic forecasts, where multiple predictive models—potentially of different types—are combined into a single aggregate predictive distribution, for example via a linear combination [3].  In such settings, the individual “experts” may output predictions in incompatible formats. For example, one model may produce a parametric distribution, while another (e.g., a deep ensemble with MC-Dropout [4]) may only provide sample-based predictions. In these cases, certain scoring rules such as the log-score cannot be applied because no unbiased estimator exists for sample-based inputs, whereas kernel scores remain applicable.
>
> This issue arises in practical domains such as economic forecasting, where individual experts may provide only point predictions or partial uncertainty information [2]. When aggregating such heterogeneous forecasts, using uncertainty measures that admit an unbiased estimator across formats becomes beneficial. We have clarified this context in the revised text.
>
> **Q2: Figure 2: You only show the AU over the land, but not over the water? Why? If so, you need to clearly specify this in the caption or at least somewhere in the text, or as an absolute minimum, somewhere in the appendix. Or does AU happen to always have the white value over the water?**
>
> Thank you for the suggestion, we did indeed miss that we did not explain this choice in the figure and added it accordingly to the description of Figure 2. We decided to show AU only for in distribution data (land), as it does not affect the out-of-distribution detection performance (as only EU is used). With that the visualization is more clear and the values of AU can be distinguished more clearly on the (purely qualitative) figure. If you believe that this should be changed, we are happy to also include the full spatial field for AU.
>
> **Q3: In line 377, what does “task loss” refer to? Is the task loss the loss that is used as training loss and as evaluation loss? This was quite confusing when reading the submission for the first time.**
>
> The task loss is essentially the training and validation loss, although it does not necessarily always have to coincide (for example one might minimize the ELBO but evaluate a model using the KL-divergence with respect to true samples. Or one might train with the robust Huber loss but evaluate a model with the RMSE). In our case, both losses coincide. We agree that this was not formulated clearly enough and changed the wording in line 377.
>
> **Q4: In Line 398, you mention “the model”. Is it actually just one individual model or an ensemble of models? If it’s an ensemble, I would write “the ensemble” instead to avoid any ambiguity.**
>
> We are sorry for the ambiguity; in fact we use an ensemble model, since otherwise we would not have access to the uncertainty measures. We adjusted the wording accordingly.
>
> **Q5: How do you suggest doing the HPO for Figure 4 in practice?**
>
> Thank you for raising this important issue. The experiment in Figure 4 was not intended to suggest tuning $\gamma$ by running parallel active-learning loops—which would indeed be impractical. Rather, it serves as an ablation to illustrate that performance can improve beyond the default median heuristic and to better understand how the kernel parameter influences the active-learning objective in spatial settings.
>
> In practice, we recommend using the median heuristic, which we employed for all main experiments in Sections 6.1 and 6.2 and which already performs well. The tuning study has been moved to Appendix C.4 and reframed explicitly as an ablation. There, we show that the optimal gamma for the T2M task is slightly larger than the heuristic value, which is consistent with the underlying spatial structure: temperatures remain correlated over relatively large distances (e.g., shared coastal regimes), making a larger bandwidth more appropriate. This analysis provides intuition for how $\gamma$ interacts with domain characteristics, but does not prescribe additional label acquisition.
>
> Following the reviewer’s suggestions, we now clearly report the median-heuristic value, interpret $\gamma$ in spatial contexts, and compare against alternative uncertainty measures (log-score, variance, CRPS). The ablation highlights potential improvements but does not affect the recommended practical pipeline, which relies on the median heuristic.

---

> ### Author Response · Authors · 2025-11-24
>
> [5/5]
>
> **Q6: [...] How practical is this guidance? Can you give a practical example where this guidance practically improves a practically relevant metric of a practically relevant downstream task?**
>
> Our recommendations are motivated by theoretical properties that offer helpful intuition for selecting between uncertainty measures, rather than prescribing universally optimal choices. For instance, for nonparametric predictive models, kernel scores admit unbiased estimators, whereas the log-score does not—making kernel-based measures more practical in settings where sampling-based estimation is required. Likewise, our robustness results show that bounded kernels should be preferred when resilience to outlier ensemble members is important.
>
> In the revised manuscript, we also provide an example linking Proposition 5.1 to downstream performance in Section 6.4, illustrating how these insights can inform uncertainty-measure selection in practice. We agree that the original phrasing may have appeared too strong and have softened the corresponding discussion.
>
> **Q7: In Line 1025-1027 of Appendix C: What is the actual input to the DRNs? Is it only the mean prediction of the ECMWF integrated (ensemble) forecast system (IFS)? Or is it actually everything that is mentioned in Section C.1? In other words, is the input only the predicted temperature or all the covariates listed in Section C.1? Does ECMWF predict all the covariates listed below or only the temperature?**
>
> The input to the DRN is the mean prediction of each of the variable mentioned below, which are all predicted by the ECMWF forecasting system. In principle one could also use only the temperature as a covariate, but in practive, post-processing seems to perform better with additional (highly correlated) covariates [Q1]. While we could in principle also use the DRN to directly predict the other covariates, we focused on T2M, as DRNs have been shown to perform well [Q1] and one can use explicit parametric assumptions and models such as deep evidential regression or deep ensembles, which might not be possible for more (non-Gaussian) covariates.
>
> **Q8: In Appendix C.1, Deep evidential regression seems to perform very poorly compared to the Deep ensemble. Do you agree?**
>
> Yes, we agree that deep evidential regression generally seems to perform worse in this scenario.
>
> **Q9: Do you see any interesting connections to optimal transport (e.g., the Wasserstein metric between probability measures)? Can you also fit them into this framework?**
>
> Thank you for the interesting question. Kernel scores (via MMD) and the Wasserstein-1 distance are indeed closely related: both are examples of integral probability metrics (IPMs) [5]. However, this class is too broad to allow for a general analysis of the induced uncertainty measures comparable to the results we obtain for proper scoring rules and, in particular, kernel scores. Our framework deliberately focuses on scoring rules because they can be expressed as expectations, admit unbiased Monte Carlo estimators, and come with well-developed theory for propriety and calibration.
>
> In addition, it has been shown that there exists no proper scoring rule whose associated divergence recovers the Wasserstein metric [6]. Consequently, Wasserstein distances cannot be embedded into our scoring-rule-based framework in a straightforward way. Designing uncertainty measures directly around optimal transport and Wasserstein distances is certainly an interesting future direction, but it lies beyond the scope of this work.
>
>
>
>
> **References:**
>
> [1] Rasp, S., Lerch, S., 2018. Neural networks for postprocessing ensemble weather forecasts. Monthly Weather Review 146, 3885–3900.
>
> [2] Krüger, F., Nolte, I., 2016. Disagreement versus uncertainty: Evidence from distribution forecasts, Journal of Banking & Finance 72, 172-186.
>
> [3] Hall, S., Mitchell, J., Combining density forecasts. International Journal of Forecasting 23, 113.
>
> [4] Gal, Y., Ghahramani, Z., n.d. Dropout as a Bayesian Approximation:  Representing Model Uncertainty in Deep Learning.
>
> [5] Müller A. Integral Probability Metrics and Their Generating Classes of Functions. Advances in Applied Probability.
>
> [6] Modeste, T., Dombry, C., 2024. Characterization of translation invariant MMD on Rd and connections with Wasserstein distances. Journal of Machine Learning Research 25, 1–39.
>
> [7] Hofman, P., Sale, Y., Hüllermeier, E., Uncertainty Quantification with Proper Scoring Rules: Adjusting Measures to Prediction Tasks, arXiv preprint arXiv:2505.22538
>
> [8] Nikita Kotelevskii, Vladimir Kondratyev, Martin Takáˇc, Eric Moulines, and Maxim Panov. From risk to uncertainty: Generating predictive uncertainty measures via bayesian estimation. In The Thirteenth International Conference on Learning Representations, 2025
>
> [9] Sale, Y., Hofman, P., Wimmer, L., Hüllermeier, E., Nagler, T.. Second-order uncertainty quantification: Variance-based measures, arXiv preprint arXiv:2401.00276

---

### Author Response · Authors · 2025-11-24
**Main revision update**

We sincerely thank all reviewers for their detailed feedback and appreciate the constructive suggestions regarding our work. In response, we expanded and strengthened the empirical evaluation, added quantitative analyses, revised sections for clarity, and incorporated several additional baselines and datasets.


**Out-of-distribution (OOD) detection:**

To move beyond the earlier qualitative assessment, we now report AUROC scores comparing epistemic uncertainty for ID vs. OOD data. For the post-processing task, we evaluate deep ensembles [1] and deep evidential regression [2]. Here, we stick to the already implemented two baselines due to computational restrictions.
Additionally, we incorporate the depth-regression benchmark from [1], using image-depth pairs and a dedicated OOD dataset. For this task, we further include deep evidential regression [1], the multivariate CRPS method [2], and a multivariate Gaussian predictor [3]. Across datasets and models, the energy-score–based measure achieves the lowest average rank. Results appear in Section 6.1 and Appendix C.1. The section is now explicitly titled Out-of-distribution detection.

**Active learning:**

We expanded the active-learning evaluation and moved it to a dedicated Section 6.4 (details in Appendix C.4). For five UCI datasets, we compare deep ensembles [1], deep evidential regression [2], CRPS-ENS [3], and mixture density networks [3]. We also revisit the T2M post-processing task [5] and evaluate all uncertainty measures under the same active-learning protocol. The earlier kernel-bandwidth tuning is reframed as an ablation and moved to the appendix; the main experiments rely on the median heuristic.
We summarize results using average ranks in Table 3. While performance varies with the task and predictor, kernel-score–based measures perform consistently well and achieve the best average rank.

**Robustness:**

Following reviewer suggestions, we now include quantitative robustness results for epistemic uncertainty in Table 2 and Appendix C.2. As expected, EU increases with stronger perturbations to individual ensemble members, but should remain stable against a single extreme outlier. The bounded-kernel score again shows the smallest change, while log- and squared-error–based measures diverge quickly. Interestingly, for EU the energy-score measure is more robust than the log-score, likely due to the geometry of the underlying divergence.

**Other changes:**

We additionally incorporated various smaller clarifications and corrections highlighted by the reviewers. All changes are marked in the revised manuscript and explained in the individual response sections.

**References**

[1] Lakshminarayanan, B., Pritzel, A., Blundell, C., 2017. Simple and Scalable Predictive Uncertainty Estimation using Deep Ensembles.

[2] Alexander Amini, Wilko Schwarting, Ava Soleimany, and Daniela Rus. Deep evidential regression. In Proceedings of the 34th International Conference on Neural Information Processing Systems.

[3] Kelen, D.M., Jung, Á., Kersch, P., Benczur, A.A., 2024. Distribution-Free Data Uncertainty for Neural Network Regression. Presented at the The Thirteenth International Conference on Learning Representations.

[4] Muschinski, T., Mayr, G.J., Simon, T., Umlauf, N., Zeileis, A., 2024. Cholesky-based multivariate Gaussian regression. Econometrics and Statistics 29, 261–281.

[5] Feik, Moritz & Lerch, Sebastian & Stühmer, Jan. (2024). Graph Neural Networks and Spatial Information Learning for Post-Processing Ensemble Weather Forecasts. 10.48550/arXiv.2407.11050.

---

### Meta-Review · Area_Chair_9s5Q · 2025-12-27

**Summary:**

While the reviewers find the results of the paper to be interesting, they have identified several major concerns, most notably regarding the presentation of the results and their positioning with respect to related and prior work, the significance and correctness of several theoretical results, and the experimental setup, including the choice of compared methods, datasets, and models. Although the authors have attempted to address many of these concerns in the rebuttal, the sheer number of required improvements and modifications far exceeds what can be reasonably handled within a short review process and would necessitate multiple rounds of back-and-forth and further evaluation. Moreover, some concerns, particularly those related to theoretical significance and novelty, appear to remain outstanding.

**Reviewer Concerns:**

As noted above, while many of the concerns appear to have been addressed in the rebuttal, several outstanding issues remain, particularly those related to theoretical soundness, significance, and novelty.

**Reviewer Scores:**

Given the magnitude of the required changes and some outstanding concerns, I believe that the reviewers who raised significant concerns and assigned low ratings would not have substantially revised their initial assessments, beyond perhaps moving to a position marginally below the acceptance threshold.

---

### Decision · Program_Chairs · 2026-01-26

Reject